# Unmanned Aircraft System (UAS) Structure-From-Motion (SfM) for Monitoring the Changed Flow Paths and Wetness in Minerotrophic Peatland Restoration

**Lauri Ikkala** [1,*]**,** **Anna-Kaisa Ronkanen** [2]**, Jari Ilmonen** [3]**, Maarit Similä** [3]**, Sakari Rehell** [3]**, Timo Kumpula** [4]**, Lassi Päkkilä** [1]**, Björn Klöve** [1] **and Hannu Marttila** [1]

[1] Water, Energy and Environmental Engineering Research Unit, Faculty of Technology, University of Oulu, P.O. Box 4300, FIN-90014 Oulu, Finland; lassi.pakkila@oulu.fi (L.P.); bjorn.klove@oulu.fi (B.K.); hannu.marttila@oulu.fi (H.M.)

[2] Finnish Environment Institute (SYKE), University of Oulu, P.O. Box 413, FI-90014 Oulu, Finland; anna-kaisa.ronkanen@syke.fi

[3] Metsähallitus Parks and Wildlife Finland, P.O. Box 94, FI-01301 Vantaa, Finland; jari.ilmonen@metsa.fi (J.I.); maarit.simila@metsa.fi (M.S.); sakari.rehell@metsa.fi (S.R.)

[4] Department of Geographical and Historical Studies, Faculty of Social Sciences and Business Studies, Joensuu Campus, University of Eastern Finland, P.O. Box 111, FI-80101 Joensuu, Finland; timo.kumpula@uef.fi

[*] Correspondence: lauri.ikkala@oulu.fi

**Abstract:** Peatland restoration aims to achieve pristine water pathway conditions to recover dispersed wetness, water quality, biodiversity and carbon sequestration. Restoration monitoring needs new methods for understanding the spatial effects of restoration in peatlands. We introduce an approach using high-resolution data produced with an unmanned aircraft system (UAS) and supported by the available light detection and ranging (LiDAR) data to reveal the hydrological impacts of elevation changes in peatlands due to restoration. The impacts were assessed by analyzing flow accumulation and the SAGA Wetness Index (SWI). UAS campaigns were implemented at two boreal minerotrophic peatland sites in degraded and restored states. Simultaneously, the control campaigns mapped pristine sites to reveal the method sensitivity of external factors. The results revealed that the data accuracy is sufficient for describing the primary elevation changes caused by excavation. The cell-wise root mean square error in elevation was on average 48 mm when two pristine UAS campaigns were compared with each other, and 98 mm when each UAS campaign was compared with the LiDAR data. Furthermore, spatial patterns of more subtle peat swelling and subsidence were found. The restorations were assessed as successful, as dispersing the flows increased the mean wetness by 2.9–6.9%, while the absolute changes at the pristine sites were 0.4–2.4%. The wetness also became more evenly distributed as the standard deviation decreased by 13–15% (a 3.1–3.6% change for pristine). The total length of the main flow routes increased by 25–37% (a 3.1–8.1% change for pristine), representing the increased dispersion and convolution of flow. The validity of the method was supported by the field-determined soil water content (SWC), which showed a statistically significant correlation ($R^2 = 0.26$–$0.42$) for the restoration sites but not for the control sites, possibly due to their upslope catchment areas being too small. Despite the uncertainties related to the heterogenic soil properties and complex groundwater interactions, we conclude the method to have potential for estimating changed flow paths and wetness following peatland restoration.

**Keywords:** wetland; rewetting; drone; UAV; spatial analysis; hydrology; surface runoff; flow accumulation; topographic wetness index; saga wetness index

## 1. Introduction

Peatland restoration has been embraced as one of the key tools for returning peatlands to their natural hydrological functions [1], safeguarding biodiversity [2], and

re-establishing carbon sequestration [3]. This is needed due to disturbances caused by drainage for land-use purposes. Peatland restoration aims to recover the surface and groundwater flow paths, i.e., redistributing water onto the peatland surface, creating conditions close to waterlogging, and thus, enabling peatland vegetation regrowth [4]. The groundwater input is essential for recovering peatland vegetation best adapted to local water quality [5]. Typical restoration actions in peatlands include damming and infilling of ditches and directing the flow away from the drainage network using embankments or by excavating auxiliary ditches [6–9]. According to discrete standpipe well measurements [10–12], the water tables typically rapidly increase after restoration. However, recovering the pristine flow paths may be impossible if the spatial nature of the ecohydrological processes is ignored. Thus, monitoring the impacts of rewetting is required to reveal the successes and failures of peatland restoration and the need for corrective actions, and to improve restoration methodologies [13]. Commonly, changes in surface elevations have occurred in peatlands after drainage due to subsidence (consolidation, compaction, shrinkage, and oxidation of peat) [14–16]. The changes in the physical structure and subsidence of peat have been reported to be strongest close to the ditches, thus potentially influencing flow paths after rewetting [15,17]. Evaluation of topographical changes in field conditions is often challenging, and new tools producing information on spatial patterns in drained and restored peatlands are needed [18]. High-resolution elevation data describes the surface in detail and would also help with technical issues common in restoration, such as planning the dimensions of dams and understanding degradation after restoration [9]. Generally, the primary (ditch infilling and dam construction) and secondary (indirectly followed by the rewetting) topographical changes caused by peatland restoration have not previously been studied using spatially high-resolution remote sensing data. Even when large-scale primary change is obvious, i.e., the ditches become infilled, a detailed investigation is needed to determine the fine-scale influences on the flow paths on the (possibly) flat surface [19]. The slow secondary change consists of peat swelling and the accumulation of new organic material, which also re-enables the hydrological buffer function of the peatland [20]. Thus, time series information of topographical changes is critical for a comprehensive understanding of peatland functionality.

The flow paths in peatlands after restoration can be visible *on-site,* but usually only during the high-water seasons. However, the flow paths can also be simulated by analyzing a digital elevation model (DEM) (e.g., [21]). The highly detailed digital surface model (DSM) and its ground-filtered derivative, digital terrain model (DTM), can be produced using airborne remote sensing [22]. Unmanned aircraft systems (UAS), in particular, allow photogrammetric mapping at a centimetre-level spatial resolution [23]. UAS mapping is an automated process providing quick results and flexibility, particularly in environments where conventional field surveys are laborious, such as wetlands [24,25]. The photogrammetric elevation model is produced using UAS data with a structure-from-motion (SfM) machine learning algorithm which combines the neighboring images and determines depth information for each pixel when the images have sufficient overlap [26]. Centimetre-level accuracy can be achieved if a high-precision Global Navigation Satellite System (GNSS) unit, such as Real-Time Kinematic (RTK), is used to follow the aircraft location or if ground control points (GCPs) with known co-ordinates are used for georeferencing [23]. Photogrammetric SfM can reach a spatial accuracy comparable with laser-based light detection and ranging (LiDAR), except in the case of densely vegetated areas where the land surface remains mostly obscured for the cameras but not for all laser beams [18]. UAS-derived DSMs and DTMs have been increasingly used to study the morphometry of peatland surfaces (e.g., [18,22,27]. Several UAS methods have also been applied in peatland restoration monitoring. Many methods [28–32] are restricted to two-dimensional products, such as orthomosaic pictures for the classification of vegetation coverage. However, there are also examples of UAS-SfM-based topographical analysis [21,33,34].

Water flows in peat are complex due to the heterogenic inner structure [35], but coarse estimates of the changed surface flow paths or flow conditions in upper acrotelm can be given by applying simple flow network analysis and topographic wetness index (TWI) based on the DTM, even without consideration of the transmissivity of peat and ground-layer vegetation such as *Sphagnum*. These algorithms were originally developed for low-resolution data [36,37]. However, the metre-class resolutions available today have shown advances in the use of ground elevations to simulate the development of soil saturation and the subsequent runoff and shallow subsurface flow (e.g., [36,38,39]. For catchments including peat cover, topography has been studied for predicting wetland distribution [40–43], optimizing hydrotopographic methodologies [37,44,45] and revealing correlations not only with hydrological variables, such as groundwater level and soil moisture [37,43,46], but also other environmental variables, such as plant species richness and soil pH [47]. De Roos et al. (2018) used an SfM-based DSM to model the hydrological one-instant flow accumulation and wetness of a temperate, fire-damaged peatland restoration site [21]. Furthermore, Dale et al. (2020) used multi-temporal SfM data to detect elevation and flow accumulation changes in tidal wetland restoration [19].

This study aimed to test and demonstrate UAS-SfM-derived DTMs to analyze the restoration success of two boreal forestry-drained minerotrophic fens by analyzing the flow paths before and after restoration. Furthermore, we estimate spatial accuracy with control data and the limitations of the topographical method from the peatland restoration aspect. We hypothesize that hydrological flow accumulation analysis and the derived topographical wetness index can reveal the impacts of changed topography in restoration, especially for major shifts, such as ditch infilling and damming. Additionally, we discuss the practical implications of UAS-based topographical analysis for future peatland restoration and monitoring actions.

## 2. Materials and Methods

### 2.1. Study Sites

We established four separate sites in two boreal regions (Mujejärvi and Olvassuo) in the southern aapa mire zone in Finland (Figure 1). The typical annual precipitation is 809 mm and 689 mm and the temperature is 2.8 °C and 1.1 °C for Mujejärvi and Olvassuo, respectively [48]. Two sites were set up for both Mujejärvi and Olvassuo: a fresh restoration site (Loukkusuo and Iso Leväniemi, respectively) and a pristine control site (Tammalampi and Kirkaslampi, respectively). The sites were selected to study the effects of restoration in areas on the edges of drained shallow peatlands and the undrained fens below them. The studied peatlands have been drained with ditch intervals of 30–35 m over recent decades (Table 1, Figure 2) to intensify tree growth for forestry. When the surface water and groundwater were channeled into the drainage network, it changed not only the ecohydrology of the ditched area but also that of the aapa mire below. Thus, the restoration affects a larger area than the ditched area alone. The regions have a history of restoration activities, and they are a part of the peatland monitoring network managed by Metsähallitus Parks and Wildlife Finland, the managing institute for the state-owned protection areas [9]. There are old restoration sites next to both pristine control sites, but the control areas have been planned so as to separate them from the hydrological impacts of the drained-restored part.

**Table 1.** Study site information.

| Region | Mujejärvi | | Olvassuo | |
|---|---|---|---|---|
| **Study Site** | **Loukkusuo** | **Tammalampi** | **Iso Leväniemi** | **Kirkaslampi** |
| Site type | Restoration | Control | Restoration | Control |

**Table 1.** *Cont.*

| Region | Mujejärvi | | Olvassuo | |
|---|---|---|---|---|
| Peatland type in the pristine state | Oligotrophic low-sedge pine fen | Oligotrophic low-sedge pine fen | Meso-eutrophic fen and flark fen | Meso-eutrophic sedge-dominated flark fen |
| **Study Site** | **Loukkusuo** | **Tammalampi** | **Iso Leväniemi** | **Kirkaslampi** |
| Drained (approximate) | 1980 | No drainage | 1970 | No drainage |
| Restored | 2020/07 | - | 2019/10 | - |
| Area of the restoration site (ha) [1] | 108.8 | - | 45.9 | - |
| Area of the watershed basin upslope (ha) | 58.8 | 18.3 | 168.3 | 12.5 |
| Area of the processing boundary (ha) | 7.6 | 7.6 | 13.8 | 8.2 |
| Mean slope inside the watershed basin upslope (%) [2] | 8.8 | 6.1 | 6.5 | 3.6 |
| Mean slope inside the processing boundary (%) [2] | 3.6 | 3.8 | 5.3 | 3.3 |

[1] According to a filtered triangulation. [2] According to the 1-metre resolution.

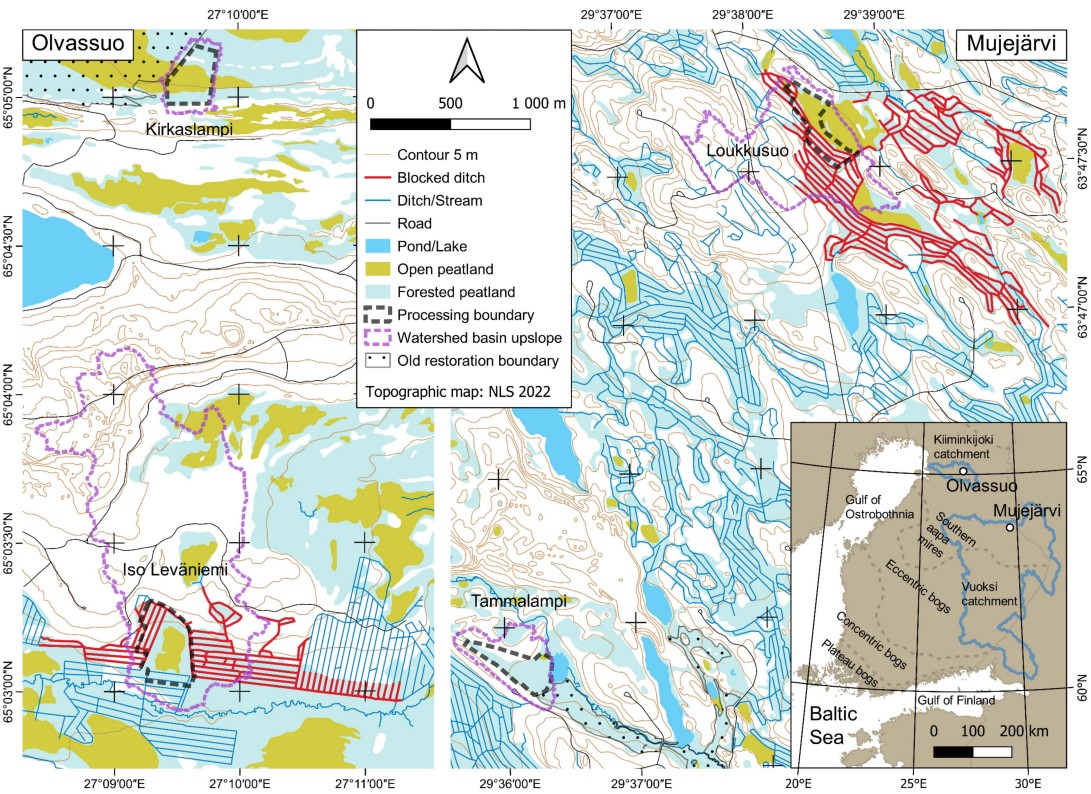

**Figure 1.** Locations of the study sites. The regions of Olvassuo and Mujejärvi are at the upper end of the catchments in the southern aapa mire zone (classification according to [49]). Loukkusuo and Iso Leväniemi are the restoration sites and Tammalampi and Kirkaslampi are their pristine control pairs, respectively. The processing boundaries do not cover the entirety of the restoration sites but focus on the edges of the downslope open peatland instead.

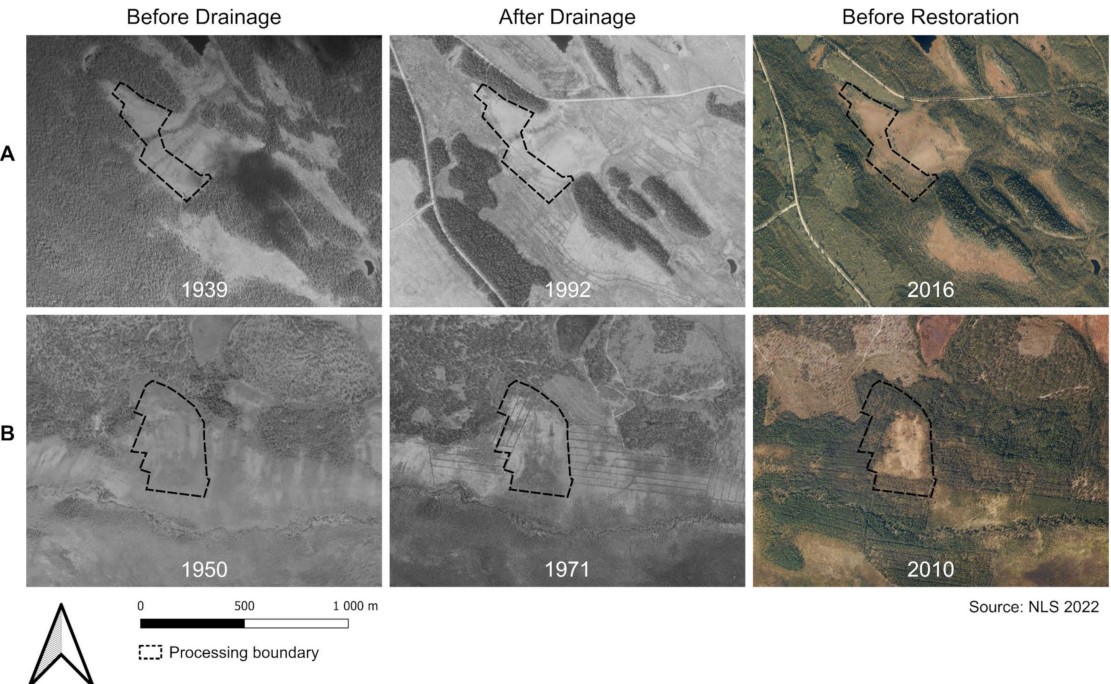

**Figure 2.** Historical development of Loukkusuo (**A**) and Iso Leväniemi (**B**) sites. The pristine states can be assessed from the aerial photos before drainage. The photos after drainage show the developing drainage networks and the first signs of forestry actions. The photos before restoration show the tree cover of the drained areas at its densest.

The Mujejärvi region consists of two conservation areas at the border of Kuhmo and Nurmes municipalities in eastern Finland. The Mujejärvi and Jonkerinsalo conservation sites, including a total of 5972 ha of coniferous forests (43% and 72%, respectively), peatlands (30% and 25%, respectively) and waterbodies (26% and 3%, respectively) [50,51]. The region is hilly due to the remaining peneplains of ancient Karelides, and the latest ice age also shaped glaci-fluvial deposits, such as drumlins and eskers [52,53]. Most of the peatlands in the region are small (a few dozen hectares). A total of 179 ha of drained peatlands in Mujejärvi have been restored since 2005. The input water for the study sites originates from soils consisting of bedrock and till (also coarse grains in Tammalampi) turning to deep peat deposits at the sites [54]. The study sites in Mujejärvi are relatively flat fens, but the surrounding hills reach to 270 m above sea level, 40 m higher than the sites. Loukkusuo is 3.8 km northeast of its pristine pair. Tammalampi is an oligotrophic low-sedge pine fen, the same type as Loukkusuo before drainage. However, the habitat was changed, and tree growth intensified in Loukkusuo after the drainage (Figure 2A).

The conservation site of Olvassuo is located at the border of the Utajärvi, Puolanka and Pudasjärvi municipalities in northern Finland. The total area protected (27,000 ha) consists mostly of peatlands (73%) and coniferous forests (20%), as well as waterbodies (4%) and mixed forests (3%) [55]. The glaci-fluvial formation of Kälväsvaara hill (200 m above sea level at its highest, 70 m higher than our sites) dominates the landscape. It is part of the esker protection program, which, however, allows for it to be used for forestry [56]. The hill covers an aquifer which discharges its waters as seepage and springs to the surrounding aapa mires of Olvassuo (on the northern side) and Leväsuo (on the southern side). The soil on the hill contains thick layers of sand, gravel, till and silt with crystalline bedrock at a depth of 50 m [57]. The large lowland fens (thousands of hectares) receive their water inputs mostly as groundwater that travels diverse (sub)surface flow paths [58]. The groundwater discharging to the sites has a higher pH than the surrounding peatlands and, thus, habitats are provided for endangered species, such as marsh saxifrage (*Saxifraga hirculus*) [55]. The sites are located in the high ends of the aapa mires, where water tables can be unstable due to the seasonally fluctuating nature of the discharge. There have been restoration activities

over a total area of 540 ha in Olvassuo since 1998. In Iso Leväniemi, the core of the UAS site was never drained, but the surrounding deep ditches cut the natural groundwater flow, meaning the water inputs for the seepage surfaces were lost. However, a significant part of the peatland remained open (Figure 2B), making UAS monitoring of the peatland surface possible. The vegetation type at the uphill end of Iso Leväniemi was open groundwater-fed meso-eutrophic fen in the pristine state. Currently, the vegetation is mostly poor *Papillosum* low-sedge fen. The lower end of the restoration site used to be a flark fen, but drainage has eradicated these species and induced the growth of tree seedlings. The location of the Kirkaslampi pristine control pair 3.3 km north of Iso Leväniemi has been optimized so that it corresponds hydrologically and geologically to the pristine state of its pair. There, the upper part has areas of meso-eutrophic fen and the lower part can be classified as meso-eutrophic sedge-dominated flark fen. The area is mostly open, except for some tree-covered strips.

The studied restorations in Loukkusuo and Iso Leväniemi were implemented in 2020 and 2019, respectively. At both sites, the ditch lines were made clearer when necessary to give room for the excavators, and the ditches were infilled with peat (or mineral soil when available) taken from unconnected pits along the ditches. Peat and geotextiles were also used to build dams (Loukkusuo) and embankments (Iso Leväniemi) using machinery (this was also performed manually in Iso Leväniemi), and small ditches were dug manually to direct water to its natural routes. No further excessive tree removal was performed even when the sites were open in their pristine states as the impact on evapotranspiration was considered low. In Iso Leväniemi, technical implementation was also limited by a short winter, preventing the use of machinery on the wet soil for a period.

### 2.2. UAS Mapping

A flight plan was set up for each of the four sites (Figures 1 and 3) in either Pix4Dcapture (for Phantom 4 Pro) or DJI Pilot (for Phantom 4 RTK). The RTK correction signal for a virtual reference station was received from the mobile network. An area of interest was chosen to represent the open and semi-open parts of the whole restoration site, as a sensible limit for the mapping area, considering operational constraints (desired ground sampling distance, flight time, battery consumption, the requirement of maintaining a visual line of sight and the post-processing performance and capacity), is typically smaller (15–20 ha with the used equipment) than the restoration implementation area (Tables 1 and 2). The UAS flights aimed to cover mutually comparable areas. However, due to the limited cover of the pristine conditions in Tammalampi, half of the mapped area covered an old restoration site from 2008.

Nine permanent GCPs were established evenly over the area of interest for georeferencing and quality control. The GCPs consisted of two boards (2 cm × 10 cm × 80 cm) painted white and attached as a cross on top of either a 1.5 m long wooden pole pushed through the peat into the mineral soil or, in the case of greater peat depths, a fresh tree stump. The poles and the stumps were cut at a level just above the ground. The GCPs were surveyed with an RTK GNSS device several times during the study project to detect possible movement. Pole anchoring (89% of the GCPs) was concluded to be stable, and the mean movements of 20 mm in the plane and 21 mm in elevation were considered to meet the accuracy of the used GNSS device. Contrarily, rising of the GCPs (a mean of 72 mm) was discovered during the restoration of Iso Leväniemi (from June 2019 to August 2020) for OI5, OI6, OI8 and OI9. The latter three were connected to stumps, but OI5 was a pole at the edge of a deep peat layer and might have been loosely anchored to the mineral soil.

The campaigns before restoration implementation are called Intervention Before (IB) and Control Before (CB), and the campaigns after restoration are called Intervention After (IA) and Control After (CA) (see Table 2). The mapping conditions varied from being fully overcast to sunlit, producing shadows in different illumination geometries. The grid-type flight plans were set as greater than the desired processing boundary to ensure a sufficient overlap for the area of interest. Altitudes of 50–90 m, together with 90% frontal and 80%

side overlaps, were set, producing centimetre-class ground sampling distances (Table 2). Each site was mapped with one flight (including changing the battery), and each restoration site and its pristine control pair were mapped on the same day. The time between the IB/CB and IA/CA campaigns was 14 months (from June to August) for Mujejärvi and 12 months (mid-August) for Olvassuo.

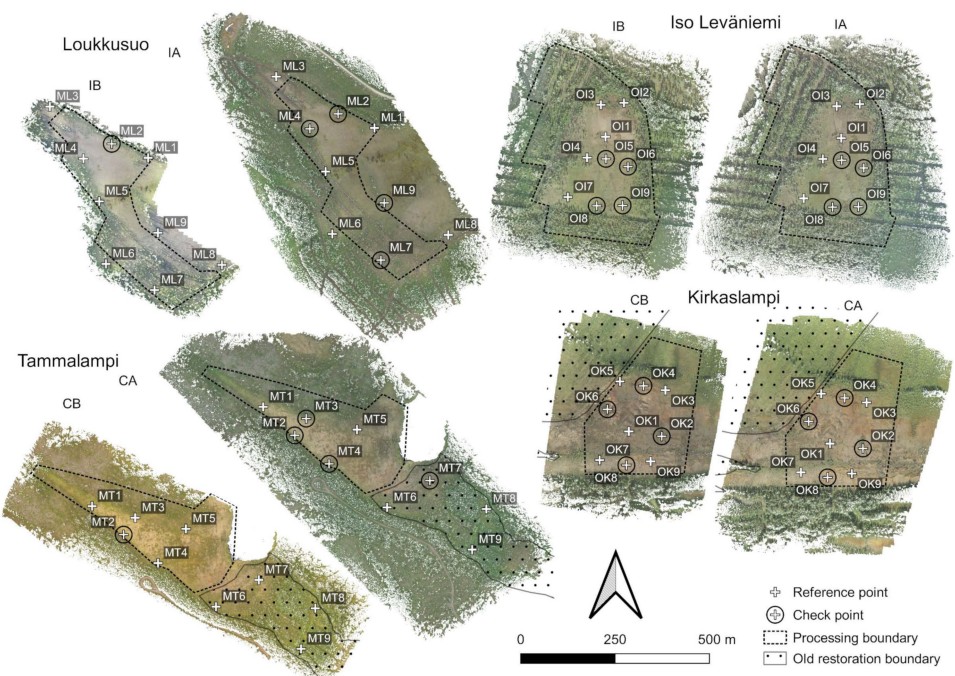

**Figure 3.** Composed orthomosaic pictures and processing boundaries. The Intervention Before (IB) and After (IA) campaigns were performed at the restoration sites and their simultaneous control campaigns (CB and CA, respectively) at the pristine sites. White crosses represent the distribution of the permanent ground control points, and the circles show the marker selection for reference (georeferencing) and control (testing the accuracy), depending on the georeferencing type (GCPs only/Real-Time Kinematic (RTK) onboard). The processing boundaries were drawn at a distance from the old restoration sites.

The campaigns before restoration implementation are called Intervention Before (IB) and Control Before (CB), and the campaigns after restoration are called Intervention After (IA) and Control After (CA) (see Table 2). The mapping conditions varied from being fully overcast to sunlit, producing shadows in different illumination geometries. The grid-type flight plans were set as greater than the desired processing boundary to ensure a sufficient overlap for the area of interest. Altitudes of 50–90 m, together with 90% frontal and 80% side overlaps, were set, producing centimetre-class ground sampling distances (Table 2). Each site was mapped with one flight (including changing the battery), and each restoration site and its pristine control pair were mapped on the same day. The time between the IB/CB and IA/CA campaigns was 14 months (from June to August) for Mujejärvi and 12 months (mid-August) for Olvassuo.

### 2.3. UAS Data Stitching

Each photo dataset from the UAS campaigns was imported into Agisoft Metashape 1.7.3 for SfM processing (Figure 4). First, the sharpness of the images was studied to find the impacts of long exposure. The Estimate Image Quality tool compares the borders of each photograph with its downscaled version to determine a value between 0–1 representing the contrast in the sharpest area of the image [59]. A campaign-specific quality threshold of 0.74–0.81 was found to reveal blur. The blur was found only on the corners of the images taken at the turns between the flight lines. Furthermore, any excess images (e.g., images of

transition lines and ground-level shots) were removed. The data reference was converted into the national ETRS89/TM35FIN (ESPG: 3067) co-ordinate system with the N2000 height grid. The photos were aligned with the bundle adjustment accuracy set to high so as to exploit the original resolution for detecting common features [60]. The neighboring images were selected according to the image co-ordinates (reference preselection "Source") and the generic preselection was used to speed up the process. Key and tie-point limits were left as defaults (40,000 and 4000, respectively) balancing the processing time and reprojection error, and adaptive camera model fitting, which would automatically select the used internal camera parameters, was not used as suggested by [61].

**Table 2.** Campaign timings and unmanned aircraft system (UAS) data and processing information according to Metashape. RMSE = root mean square error.

| Site | Loukkusuo | | Iso Leväniemi | | Tammalampi | | Kirkaslampi | |
|---|---|---|---|---|---|---|---|---|
| Site Type | Restoration | | Restoration | | Control | | Control | |
| Campaign Type | IB | IA | IB | IA | CB | CA | CB | CA |
| Timing of campaign | 24/6/2019 | 18/82020 | 20/8/2019 | 21/8/2020 | 24/6/2019 | 18/8/2020 | 20/8/2019 | 21/8/2020 |
| Aircraft | Phantom 4 Pro | Phantom 4 RTK | Phantom 4 RTK | Phantom 4 RTK | Phantom 4 Pro | Phantom 4 RTK | Phantom 4 RTK | Phantom 4 RTK |
| Number of aligned cameras [1] | 769 | 749 | 439 | 438 | 672 | 832 | 296 | 339 |
| Flying altitude [2] (m) | 50 | 96 | 113 | 124 | 91 | 100 | 105 | 117 |
| Ground resolution (cm/pixel) | 1.24 | 2.38 | 2.81 | 3.08 | 3.33 | 2.50 | 2.58 | 2.89 |
| Coverage (ha) | 12.4 | 32.4 | 25.6 | 29.2 | 23.7 | 35.0 | 20.0 | 27.4 |
| Number of tie-points | 269,289 | 155,407 | 116,245 | 108,788 | 94,576 | 142,383 | 81,355 | 89,035 |
| Number of projections | 1,541,084 | 1,294,303 | 675,735 | 667,877 | 916,388 | 1,117,907 | 793,089 | 823,815 |
| RMSE of normalized reprojection (pixels) | 0.498 | 0.459 | 0.407 | 0.427 | 0.507 | 0.444 | 0.411 | 0.422 |
| Average tie-point multiplicity | 4.43 | 5.98 | 4.09 | 4.27 | 5.97 | 5.74 | 7.84 | 7.24 |
| Timing of reference campaign | 17/6/2020 | - | 17/6/2018 | - | 17/6/2020 | 17/6/2020 | 20/6/2018 and 19/8/2015 [3] | 20/6/2018 and 19/8/2015 [3] |
| Number of soil water content samples | - | 17 | - | 25 | - | 16 | - | 25 |

[1] Number of images that were successfully aligned. [2] Campaign mean distance between the cameras and the sparse cloud. [3] The dataset includes data from two campaigns and the exact border is unknown.

The GCP markers were assigned and either manually refined for all found projections or removed when visually ambiguous. For precise non-RTK (i.e., RTK not onboard) mapping, georeferencing with GCPs is inevitable [23,62]. Despite direct georeferencing being possible with the onboard-RTK, the use of some GCPs is recommended for full accuracy. In the case of small square mapping extents, at least four GCPs are recommended [63]. For larger extents, at least one additional GCP is recommended in the central area [64,65]. Eight GCPs were used in this study for georeferencing the non-RTK campaigns (Muje-

järvi IB/CB) and the camera locations, together with five GCPs for the RTK campaigns (Figure 2). Correspondingly, the quality of the model was tested by using one or four GCPs as checkpoints for non-RTK and RTK campaigns, respectively. The reference point (the Metashape term 'control point' is not used in this paper to prevent confusion with validation approaches) and checkpoint selection should be based on even distribution [66]. The reference points of the RTK campaigns in Loukkusuo and Tammalampi were chosen from the middle and ends of the elongated flight plans. In Iso Leväniemi, the selection was defined using the firmly anchored pole-points, despite their poor (linear) geometry. Only the GCPs in Kirkaslampi were distributed as a systematic grid, permitting a quincunx shape for the network. For the non-RTK-campaigns, the GCP closest to its neighbors (i.e., at the highest GCP density) was selected as the only checkpoint. According to the metadata, the mean accuracy for the RTK-measured camera locations was 22 mm, while a constant 20 mm accuracy was assumed and set for the ground-surveyed reference points to give appropriate weight for georeferencing.

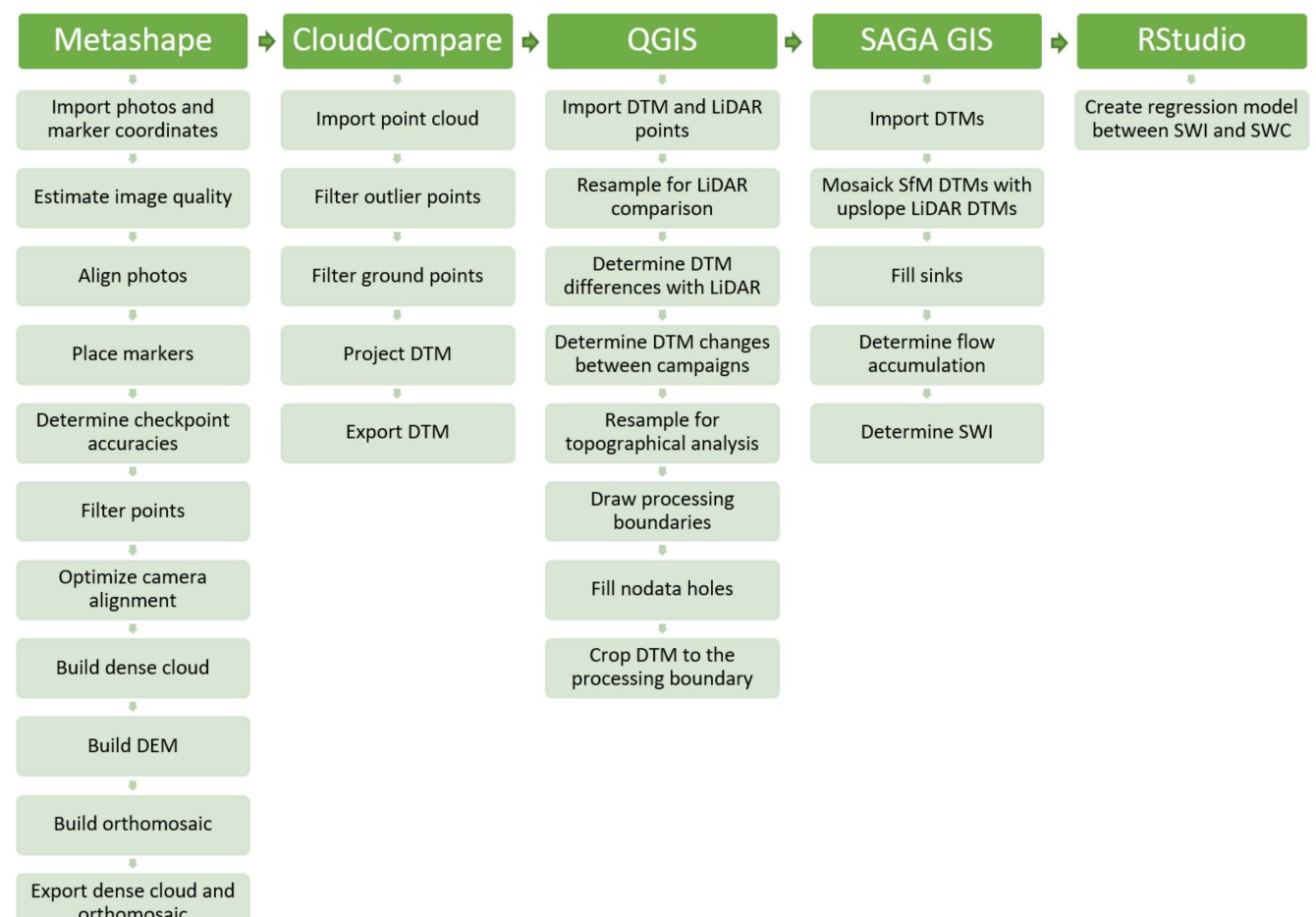

**Figure 4.** The data processing workflow of the software. The steps were applied for the data from restoration sites to reveal the impacts of ditch infilling and damming, and for the data from pristine control sites for the impacts of uncertainties.

Poor tie-points were filtered out with the Gradual Selection tool (Figure 5A) by allowing the removal of no more than 10–20 % of the total tie-points in each step. This would avoid over-constraining and the following doming deformations [67]. Camera optimization was performed before filtration and after each filter step using the reference points and, for the RTK campaigns, the camera locations. To balance the representation of the vertical structure (for applications outside this paper) and noise level, we produced a dense cloud using moderate depth filtering at high quality. The parametrization of the

Metashape filters can be found in Supplementary Materials, Table S1. Finally, a DEM and an orthomosaic picture were produced for each dataset from the Dense Cloud with the resolutions suggested by the software (Table 2) without interpolation and hole-filling. The Dense Cloud was exported to an LAZ format for further processing and the orthomosaic was exported to a TIF format.

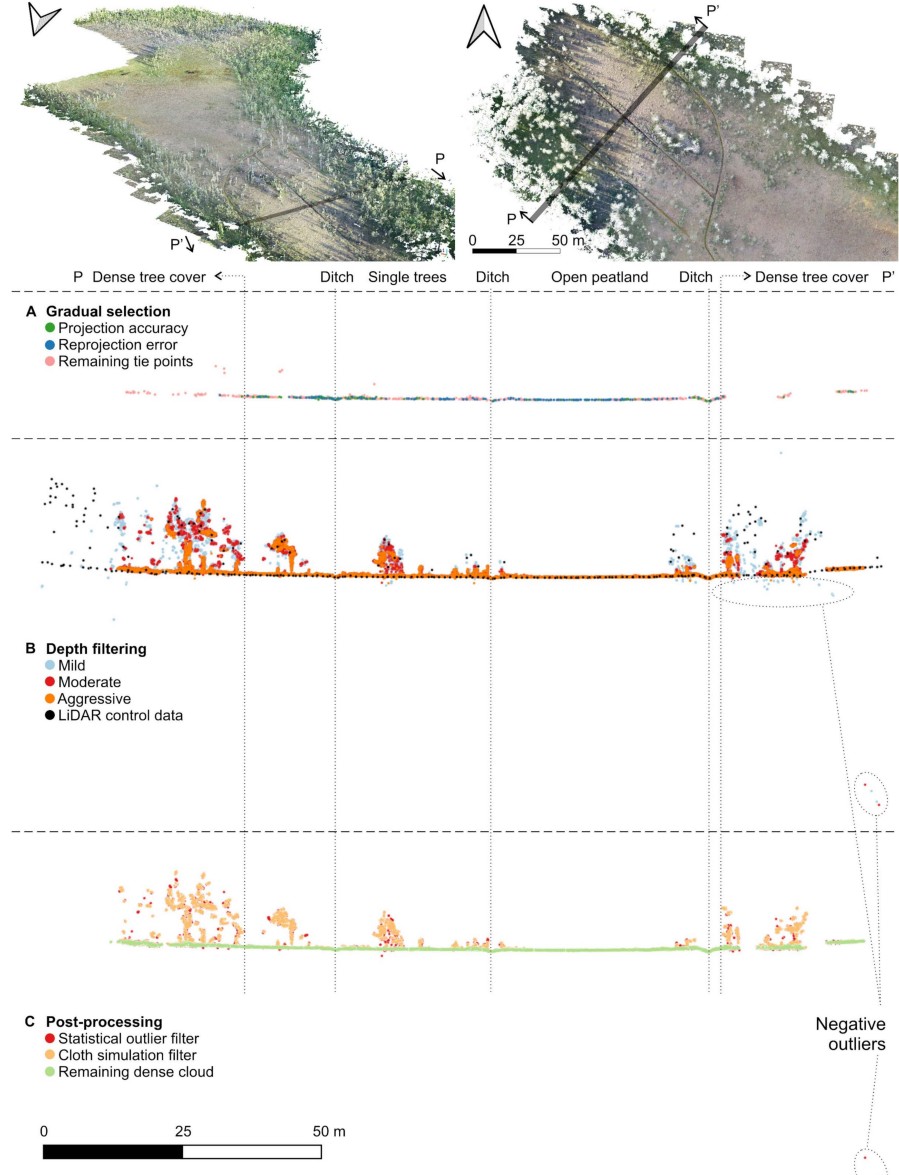

**Figure 5.** Impacts of different filtrations on the structure-from-motion (SfM) point clouds are visualized as orthometric crosscuts (3-metre wide slices for sufficiently accumulating points) for Loukkusuo site data. The slices with the fewest points are ordered closest to the viewer. The Gradual selection filters of Metashape (**A**) remove the tie-points of poor quality (the reconstruction uncertainty filter did not influence the chosen slice). The level of depth filtration in the Metashape dense cloud creation (**B**) affects the vertical representation of the vegetation and the occurrence of outliers (the light detection and ranging (LiDAR) data shown as a reference). The post-processing filtrations applied in CloudCompare for the moderately-depth-filtered dense cloud (**C**) show Statistical Outlier Removal (SOR) performing well, removing the outliers, and Cloth Simulation Filter (CSF) removing the non-ground points. (**A**,**C**) show the incremental removal of points for the same dataset, whereas (**B**) compares the products from different settings of parallel processes. The scale bar represents both horizontal and vertical directions.

### 2.4. Noise Removal and Extraction of the Terrain Model

Each dense cloud was imported to open-source CloudCompare 2.12 alpha for cleaning and classifying the points (see an example in Figure 5C) anticipating the topographical analysis. SfM point clouds typically have a high noise level, and outliers exist due to, for example, complex geometric structures and reflections [68]. Despite the filtrations performed in Metashape, three-dimensional (3D) inspections revealed both positive and negative outliers situated above and below the mapped surface, respectively [69]. The Statistical Outlier Removal (SOR) tool [70] was first used for selecting and removing the points far from real surfaces. We selected the tool parameters based on the literature (see Section S2 in the Supplementary material) and visual assessment of the iterated products. A k = 50 together with an nSigma = 0.5 were found to be optimal for cleaning the dataset. Some obvious negative outliers related to reflections from water were visually recognized after filtration and were removed by hand.

Furthermore, a Cloth Simulation Filter (CSF) was applied to classify the ground points. We were not able to find any CSF studies of peatlands, but parametrizations for other environments are available (see Section S3 in the Supplementary material). We chose the following values for the parameters, based on the literature and our experiments: scenes = flat, slope processing = true, cloth resolution = 1.0 m, classification threshold = 0.1 m and the maximum number of iterations = 500.

The remaining ground points were projected into the plane and exported to TIF format with a rasterize tool, selecting a grid size of 0.1 m. The cell heights were calculated as averages without interpolation. Similarly, rasterization and exportation were performed in CloudCompare for the control dataset—pre-classified LiDAR data from the National Land Survey of Finland (NLS) (see Section 2.5 for data details). As well as the rasterized 2 m TIF, the ground-filtered LiDAR points were also exported as ASCII points.

### 2.5. Evaluation of the Terrain Model

The DTM accuracy was studied using three approaches (a in Metashape, b–c in QGIS 3.20.3-Odense) using the root mean square error (RMSE) relative to the reference data with spatial comparisons. Firstly, for (a), the SfM-modelled locations of the checkpoint markers were compared with the field-surveyed RTK co-ordinates. In the second approach (b), the SfM-derived ground elevations (IB/CB/CA state) were compared with the control LiDAR data downloaded from the Open Data File Download Service by NLS. The LiDAR data (LAZ format) with a point density of 0.5 points/$m^2$ was produced between 2015–2020 from airborne campaigns. The horizontal and vertical accuracies were 60 cm and 15 cm, respectively (45 cm and 10 cm, respectively, for data from 2020 onwards) for unambiguous objects. Although the resolution of the LiDAR data corresponds to approximately 1.4 m, we found this sampling distance to be too long, considering the 45–60 cm horizontal accuracy. Thus, we decided to resample the elevations of the 10 cm UAS-derived DTM to 50 cm with the warp (Reproject) tool as averages and compared the ground elevation of each LiDAR ASCII point with the 50 cm UAS grid. For visualization, the point-wise comparison was rasterized into a 2 m grid. In the third approach (c), the combined effect of the mapping errors and natural surface variation (i.e., the development of phenology and possible mire breathing) was estimated by comparing the CB and CA SfM elevations (resolution of 10 cm) at the pristine control sites, where no anthropogenic surface changes had occurred. Finally, a level of detection (LoD) was calculated for approaches a-c to quantify the sufficient accuracy for elevation change detection as follows [19,71]:

$$\mathrm{LoD} = t\left(\sigma_{Z1}^2 + \sigma_{Z2}^2\right)^{\frac{1}{2}} \tag{1}$$

where *t* is the required level of confidence (for 95%, *t* = 1.96 was used) and $\sigma_{Z1}$ and $\sigma_{Z2}$ are the involved $\mathrm{RMSE}_Z$ values.

## 2.6. Topographical Analysis

The DTM differences between the IB and IA states at the restoration sites were determined by comparing the elevations (resolution of 0.1 m) in QGIS. Processing borders for the topographical analysis were drawn considering the data quality and assumed flow routing. The poor-accuracy data on the DTM edges were excluded according to a visual assessment (see Figure 6 in the Section 3). For the pristine sites, a minimum distance of eight metres to the neighboring old restoration sites was required. The induced effects of the elevation changes on the flow accumulation and soil moisture conditions were estimated using a workflow in QGIS and the open-source software SAGA GIS 7.8.2, known as a flexible and comprehensive software for topographical analysis [72].

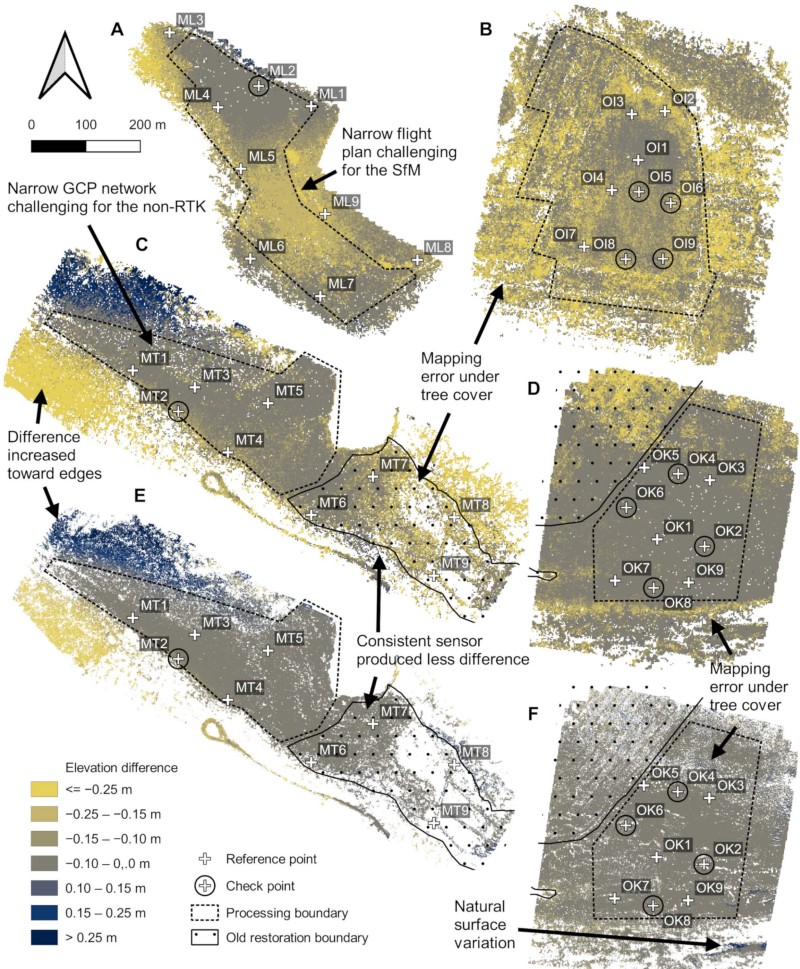

**Figure 6.** Ground elevation accuracy according to the differences between SfM and the controlling LiDAR data before the restoration. (**A**–**D**), positive values correspond to the LiDAR surface being higher and between two SfM campaigns at the pristine control site. (**E**,**F**), positive values correspond to the ground surface elevated by time in Loukkusuo (**A**), Iso Leväniemi (**B**), Tammalampi (**C**,**E**) and Kirkaslampi (**D**,**F**). The SfM campaigns of (**A**,**C**) and the CB campaign of (**E**) have been georeferenced with the GCPs only, which can be seen as decreasing accuracy as the distance from the GCP network increases. The rest of the SfM campaigns have been georeferenced using the onboard-RTK data together with five GCPs, which resulted in a smoother accuracy distribution (**B**,**D**,**F**). The checkpoints in (**E**) represent the CB campaign.

The concept of TWI is based on the theory that the tendency of a specific terrain location to accumulate water is defined by the water input (represented by the catchment area upslope) and the water output (represented by the local slope). The primal index [73]

was defined as $\ln(A_S/\tan\beta)$, where $A_S$ is the specific catchment area draining through the location and $\tan\beta$ is the local slope. The specific catchment area corresponds to the total contributing upslope area divided by the flow width, i.e., a finite distance under observation [74], the width of the grid cell in our case. We used the contributing area values, i.e., the flow accumulation map, to reveal the changes in the surface flow paths due to the restoration.

The original TWI is known for poorly describing flat sites where the local wetness is mostly driven by variables other than topography [72]. On even surfaces, the simulated flow paths are also more sensitive to elevation uncertainty [41]. To develop the methodology, [75] released the SAGA wetness index (SWI), which uses the multiple-flow algorithm presented by [76] adapted to account for high water tables typical in flat areas. A flow accumulation algorithm regulates how each package of water created by constant rainfall for each grid cell moves toward the downslope [45]. According to Freeman et al. (1991), the flow is distributed to all downslope cells with a convergence of 1.1 [76]. The values of SWI are calculated as for TWI, but the specific catchment area is iteratively modified according to the maximum specific catchment area in the immediate area:

$$A_{SM} = A_{Smax} \left(\frac{1}{t}\right)^{\beta \exp(t^\beta)} \text{ for } A_S < A_{Smax} \left(\frac{1}{t}\right)^{\beta \exp(t^\beta)} \tag{2}$$

$$\text{SWI} = \ln\left(\frac{A_{SM}}{\tan\beta}\right) \tag{3}$$

where $A_{SM}$ is the modified catchment area, $A_{Smax}$ is the maximum specific catchment area in the neighboring cells, $t$ is the suction parameter and $\beta$ is the slope angle. The suction effect means the capillary attraction in the neighboring soil to void [37]. SWI has shown promising results describing the soil moisture distribution, particularly when used with fine-scale (~1 m) data [37,46]. A resolution of 1 m was chosen for the analysis, as it is still able to describe the ditches, dams and infillings, while finding a balance between the representation of microtopography and the formation of realistic hydraulic gradients. Since cell aggregation includes a risk of diminishing the drainage features [36], the minimum resampling method was used in the warp (Reproject) tool in QGIS. The resampled DTMs were then clipped according to the processing boundaries.

The ubiquitous 2 m LiDAR DTM from the IB/CB state was used for estimating the water input from the catchment upslope. The holes in the LiDAR and SfM DTMs were interpolated by inverse distance weighting with the Fill nodata tool in QGIS, using a maximum search distance of 3 m. The LiDAR DTM elevations inside the processing boundary were replaced with SfM data—IB, IA, CB and CA datasets, separately—using the Mosaicking module in SAGA GIS and unifying the resolution to 1 m with the nearest neighbor resampling. The pits and hollows in the DTM need to be either filled or breached to allow the flow routing algorithm to access a lower cell [77]. We chose to fill the sinks with an efficient priority-flood algorithm introduced by [78]. A minimum slope of 0.01° was applied in the filling to maintain flow and prevent division by zero. The tool also delineated the watershed basins that are described in Table 1 and shown in Figure 1. The SAGA wetness index module was used with the filled DTM as the elevation dataset, a suction of 256, the catchment area type as specific, the slope type as catchment, a minimum slope of 0, an offset slope of 0.1 and a slope weighting of 1. The flow accumulation raster from the index calculation was also vectorized with the channel network module with an initiation threshold of >1000 and a minimum segment length of 100 m to enable the comparison of the main flow routes.

Besides showing the changed flow paths and water accumulation in restoration, the same topographical analysis was performed for the CB and CA datasets from the pristine control sites to show the sensitivity of the methodology for external factors, such as developing phenology and error sources in production.

### 2.7. Regression of Field Measurements on the Predicted Wetness

A validation dataset for the SWI was produced by collecting 17–25 (Table 2) surface soil water content samples from each site on the day of the IA/CA UAS campaign. Approximately 1–2 dl of the surface matter, mostly consisting of *Sphagnum*-dominated biomass, was collected. The location of each sample was recorded with a RTK GNSS device. The samples were weighed in the laboratory as wet and then again after being dried in an oven set to 60 °C. The weight difference revealed the gravimetric soil water content (SWC). The noise observed in the SWI map was smoothed with the Resampling Filter module in SAGA GIS with a scale factor of 2. The topographical wetness was then extracted from the SWI map for each SWC sampling point. The tool first averages the values by aggregation and then recovers the original resolution using spline interpolation. Eventually, linear regression models were created between the field-measured SWC and the topography-based SWI in RStudio 1.4.1106.

## 3. Results

### 3.1. Structure-From-Motion Accuracy

The field-surveyed GCP co-ordinates showed the UAS-derived surface to model the terrain at a centimetre-level accuracy (Table 3a). The checkpoints reached a mean RMSE of 20 mm on the plane and 16 mm in elevation when excluding the IA campaign at Iso Leväniemi, where movement of the unstable checkpoint GCPs during the monitoring period was observed. Comparing the ground filtered SfM DTM with the LiDAR elevations (Table 3b) showed a mean $RMSE_Z$ of 153 mm for all datasets, including dense tree cover and areas with low image overlap or ones far away from the GCP network. The processing boundaries were drawn excluding the poor-accuracy edges (Figure 6) and thus, the accuracy of the data included in the topographical analysis was refined down to a mean $RMSE_Z$ of 98 mm. Figure 6A–D reveals that the LiDAR ground surface mostly lies lower than the photogrammetric SfM elevations. When comparing two SfM elevation datasets from the pristine control sites to each other (Table 3, Figure 6E–F, more details in Table 4a), the overall mean $RMSE_Z$ of 101 mm was narrowed down to 48 mm after the exclusion of the edges. The similarity was seen to increase when a consistent sensor or georeferencing type was used for production. In Kirkaslampi, where two onboard-RTK SfM datasets were compared, $RMSE_Z$ was 40% smaller than when comparing one with the LiDAR dataset. Kirkaslampi had the lowest $RMSE_Z$ of 99 mm for the whole area. A treeless sample extent in the centre showed the SfM surface to be on average 21 mm lower than the LiDAR surface, while it was 87 mm lower for the tree-covered strip north of the GCPs. In Tammalampi, where the onboard-RTK co-ordinates were available for the CA campaign only, the discrepancy increased with the increasing distance from the GCP network, for the tree-covered areas especially. Similarly, the determined LoDs depended on the control data type. The checkpoints suggested that the smallest elevation differences able to be distinguished were on average ±52 mm, excluding Iso Leväniemi. The mean LoD for the pristine cell-wise SfM comparison was ±133 mm and for the LiDAR comparison was ±310 mm.

### 3.2. Topographical Changes Due to the Restoration

The mean elevation increase for the elevated ground was 158 mm and the mean subsidence for the subsided ground was 202 mm. The major (primary) changes included the impacts of excavation raising or lowering the ground along the ditch lines, approximately 0.6 m and 1.0 m for Loukkusuo and Iso Leväniemi, respectively. This can be seen in Figure 7A1,A3,B1. Constructed dams were also recognized, as were relevant no-changes, such as discontinued infillings due to excavators being unable to work on the wet slushy soil (such as in Figure 7A2). Minor secondary peat swelling was found in Loukkusuo at the northern edge (Figure 7A). Despite most of this rise being located in the zone of <10 cm accuracy, the checkpoint ML2 in the area showed a rise of 46 mm (compared with $RMSE_Z$ of 28 mm for the dataset). Thus, the observed swelling is probably due to inaccurate SfM

modelling. In Iso Leväniemi, the observations of swelling seemed reliable and focused on the lower surfaces, such as in Figure 7B2,B3. Secondary subsidence was mainly located in the areas of poor accuracy under dense tree covers, such as at the western edges of Figure 7A,B. Northeast from A3, the IB SfM elevation was on average 191 mm higher than the LiDAR elevation and the SfM elevation in the corresponding area decreased by 95 mm after the restoration, i.e., the non-RTK-derived SfM surface approached the elevation where it was supposed to be, according to the control data. Thus, it remains unclear if the change was due to restoration or surface modelling errors.

### 3.3. Changes in Flow Accumulation and Wetness

The topographical analysis showed changes in the flow paths and wetness due to restoration (Figure 8A,B). The water flows were shifted away from the ditch bottoms more evenly onto the peatland surface in Loukkusuo where the ditches were dug perpendicular to the general hydraulic gradient (Figure 8A). The drainage network failed to hold the flow in the ditches in the IB state as it previously discharged through three main routes in the undrained part. Nonetheless, restoration distributed the same flow input to >10 routes and the change was shown through the total length of the main routes, which increased by 37% after restoration (Table 4b). Redistribution also caused changes in the SWI values of Loukkusuo. In addition to the former ditch-bottoms becoming dryer and earlier slopes wetter, the area between and below the infilled functioning ditches also became wetter at the price of the drying surroundings of the earlier main routes on the undrained section. The overall movement towards wetter conditions can be seen in the SWI histogram in Figure 8A and Table 4c. Mean SWI increased by 2.9%, while STD decreased by 15%, representing less spatial variation in wetness conditions after the restoration. According to the SWI, Loukkusuo was wetter than its control pair in the degraded state and, after the restoration, it had less variation in wetness.

**Table 3.** The georeferencing accuracy of the UAS data for the Intervention Before (IB) and After (IA) campaigns at the restoration sites, and their control campaigns (CB and CA) at the pristine sites from three aspects. The field-surveyed checkpoint co-ordinates were compared with those modelled in the SfM process (a). Sparse airborne LiDAR control elevations were compared pointwise with the SfM data (b). Rasterized elevations from two UAS campaigns were compared with each other at the sites where no anthropogenic structure changes occurred (c).

| Site | Loukkusuo | | Iso Leväniemi | | Tammalampi | | Kirkaslampi | |
|---|---|---|---|---|---|---|---|---|
| Site type | Restoration | | Restoration | | Control | | Control | |
| Campaign | IB | IA | IB | IA | CB | CA | CB | CA |
| (a) UAS checkpoint (mm) -RMSE$_{XY}$ -RMSE$_{Z}$ | 30.6 [1] 1.6 [1] | 25.8 27.8 | 16.5 17.7 | 51.1 [1] 46.9 [1] | 8.4 [2] 1.8 [2] | 38.2 20.8 | 11.7 27.6 | 11.2 12.1 |
| -LoD | ±54.6 | | ±98.3 | | ±40.9 | | ±59.1 | |
| (b) LiDAR control data (mm) -Whole dataset RMSE$_{Z}$ -Analysis area RMSE$_{Z}$ | 148.4 134.1 | - - | 207.8 173.4 | - - | 222.0 86.6 | 145.9 85.0 | 97.8 54.2 | 98.9 53.0 |
| -Analysis area LoD | ±371.7 [3] | | ±480.6 [3] | | ±237.8 | | ±148.6 | |
| (c) Pristine UAS temporal (mm) -Whole dataset RMSE$_{Z}$ -Analysis area RMSE$_{Z}$ | - - | | - - | | 141.5 56.1 | | 59.8 40.2 | |
| -Analysis area LoD [3] | - | | - | | ±155.5 [3] | | ±111.4 [3] | |

[1] Checkpoints moved due to loose anchoring. [2] One checkpoint was used (instead of the four used for the others). [3] $\sigma_{Z1} = \sigma_{Z2}$ was assumed.

**Table 4.** Results of the topographic analysis such as the development in elevation (a), flow routes (b) and Saga Wetness Index (SWI) (c) and the site-specific scaling of the field-measured Soil Water Content (SWC) distribution (d). The changes represent restoration impacts for the IB-IA and DTM uncertainties for the CB-CA comparisons.

| Site | Loukkusuo | | | Iso Leväniemi | | | Tammalampi | | | Kirkaslampi | | |
|---|---|---|---|---|---|---|---|---|---|---|---|---|
| Site Type | Restoration | | | Restoration | | | Control | | | Control | | |
| Campaign | IB | IA | Change | IB | IA | Change | CB | CA | Change | CB | CA | Change |
| (a) Cell statistics of the significant elevation changes [1] | - | - | 0.598 | - | - | 1.152 | - | - | 0.313 | - | - | 0.087 |
| -Area elevated (ha) | - | - | 7.9 | - | - | 8.3 | - | - | 4.1 | - | - | 1.1 |
| -Area elevated (%) [2] | - | - | 149.7 | - | - | 166.7 | - | - | 130.0 | - | - | 124.4 |
| -Mean rise (mm) | - | - | 64.7 | - | - | 95.6 | - | - | 31.4 | - | - | 26.2 |
| -STD rise (m) [3] | - | - | 0.720 | - | - | 0.808 | - | - | 0.128 | - | - | 0.036 |
| -Area subsided (ha) | - | - | 9.5 | - | - | 5.9 | - | - | 1.7 | - | - | 0.4 |
| -Area subsided (%) [2] | - | - | 151.7 | - | - | 252.5 | - | - | 119.2 | - | - | 123.9 |
| -Mean subsidence (mm) | - | - | 62.1 | - | - | 172.3 | - | - | 25.3 | - | - | 29.5 |
| -STD subsidence (mm) | | | | | | | | | | | | |
| (b) Total length of the main routes, proportional to the area of the processing boundary and the change (%) [4] -Total length (m) | 2612 | 3566 | +36.5 | 4775 | 5983 | +25.3 | 1910 | 1755 | −8.1 | 2914 | 3019 | +3.1 |
| -Divided by area (m/m²) | 344 | 469 | +36.5 | 346 | 434 | +25.3 | 251 | 231 | 8.1 | 355 | 368 | +3.1 |
| (c) SWI cell statistics and the change (%) [4] -Mean | 10.55 | 10.86 | +2.9 | 8.95 | 9.57 | +6.9 | 9.81 | 10.05 | +2.4 | 9.97 | 9.93 | −0.4 |
| -STD | 2.52 | 2.14 | −15.1 | 2.95 | 2.58 | −12.5 | 2.23 | 2.31 | +3,6 | 1.91 | 1.85 | −3.1 |
| (d) SWC extreme values (m%) [5] -SWC$_{min}$ | - | 192 | - | - | −158 | - | - | 660 | - | - | 1479 | - |
| -SWC$_{max}$ | - | 2143 | - | - | 3728 | - | - | 1266 | - | - | 1818 | - |

[1] According to the 100-mm DTM cropped to the processing boundary. Significant change > 100 mm. [2] In relation to the whole area. [3] Standard deviation (STD). [4] The change between the two states. [5] According to the linear regression model (Supplementary Materials Figure S1). SWC$_{min}$ = SWC (SWI = 0). SWC$_{max}$ = SWC (SWI = 20).

### 3.4. Changes in Flow Accumulation and Wetness

The topographical analysis showed changes in the flow paths and wetness due to restoration (Figure 8A,B). The water flows were shifted away from the ditch bottoms more evenly onto the peatland surface in Loukkusuo where the ditches were dug perpendicular to the general hydraulic gradient (Figure 8A). The drainage network failed to hold the flow in the ditches in the IB state as it previously discharged through three main routes in the undrained part. Nonetheless, restoration distributed the same flow input to >10 routes and the change was shown through the total length of the main routes, which increased by 37% after restoration (Table 4b). Redistribution also caused changes in the SWI values of Loukkusuo. In addition to the former ditch-bottoms becoming dryer and earlier slopes wetter, the area between and below the infilled functioning ditches also became wetter at the price of the drying surroundings of the earlier main routes on the undrained section. The overall movement towards wetter conditions can be seen in the SWI histogram in Figure 8A and Table 4c. Mean SWI increased by 2.9%, while STD decreased by 15%, representing less spatial variation in wetness conditions after the restoration. According to the SWI, Loukkusuo was wetter than its control pair in the degraded state and, after the restoration, it had less variation in wetness.

The effects at the more sloped site of Iso Leväniemi were milder, since most of the mapped ditches were directed downhill, and thus, the flow tended to follow the same routes, despite the restoration (Figure 8B). However, the convolution of the flow increased due to the ditch infillings leaving roughness on the surface. Furthermore, the dispersion

of the flow in the not-ditched central part increased. The total length of the main routes, representing both flow dispersion and convolution in this case, increased by 25% (Table 4b). The convolution was present in the IB state for the flow route at the eastern edge, likely due to the dense understory vegetation skewing the ground surface. The unfilled ditches outside the implementation area below the site continued gathering the flow after the restoration. Similar to Loukkusuo, a shift toward higher SWI values was found in the histogram, with an increase of 6.9% in mean and a decrease of 13% in STD (Figure 8B, Table 4c). Levelling the slopes of the infilled ditches increased the SWI values, despite remaining as flow routes. However, most of the area between the ditches remained dry as these strips were parallel with the overall hydraulic gradient. For swelled areas, no clear trend in wetness conditions was found. The development depended more on the changed flow paths than on the surface rise itself. SWI predicted the wetness in Iso Leväniemi to be approaching, but not reaching, the level of the control pair. Pristine Kirkaslampi still had less variation in wetness, even after the restoration.

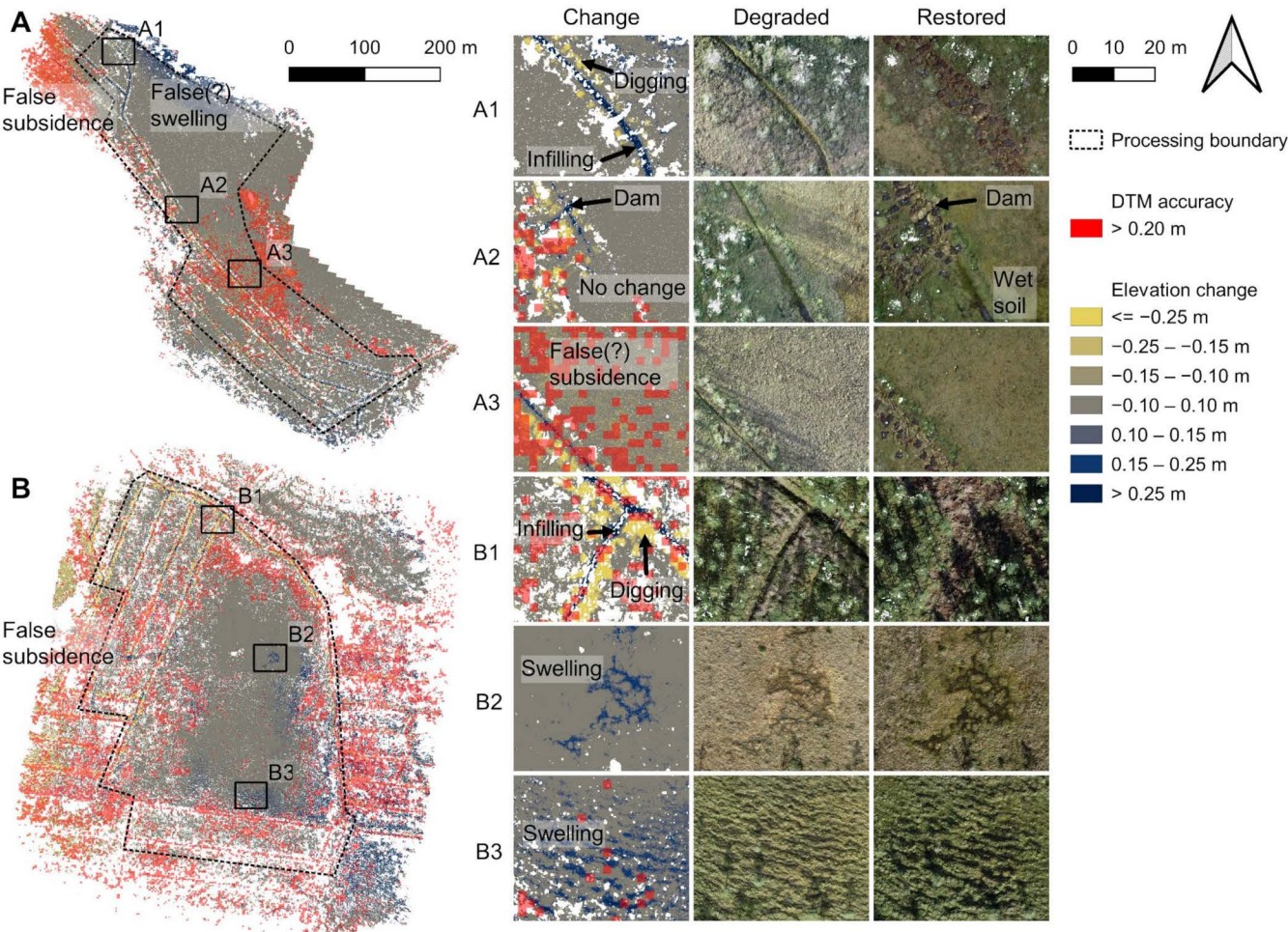

**Figure 7.** Topographic and visual changes in the restoration of Loukkusuo (**A**) and Iso Leväniemi (**B**) according to the before (IB) and after (IA) SfM datasets. Positive elevation changes correspond to surface rise after restoration. The insets (**A1–B3)** show examples of changed areas in detail and the orthomosaics of the IB and IA campaigns. The topography has been emphasized by overlaying a hillshade Digital Terrain Model (DTM) on top of the orthomosaic picture. Any elevation inspections in the areas of poor accuracy (error > 20 cm according to the reference LiDAR data, coloured as semitransparent red) should be performed with caution.

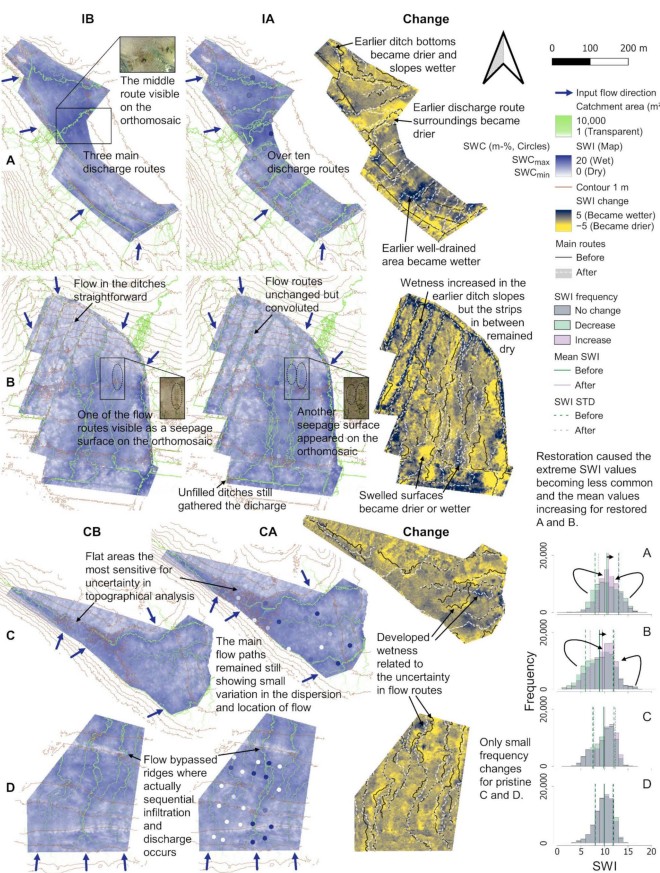

**Figure 8.** Wetness and flow path changes between the IB and IA campaigns in Loukkusuo (**A**) and Iso Leväniemi (**B**) and between the CB and CA campaigns in Tammalampi (**C**) and Kirkaslampi (**D**). The flow accumulation map (i.e., the catchment area upslope for each pixel) is drawn on top of the SWI map. A similar water input (upslope flow accumulation) to the processing boundary was assumed for both states. The changes are assumed to be caused by restoration (for **A**,**B**) or mapping errors and natural changes in the surface (for **C**,**D**). The histograms include SWI distributions for both states so that the frequency changes between the two campaigns can be elaborated from no-change. The circles fitting well with the surrounding wetness map represent a good correlation between the topography-predicted SWI and the field-measured SWC (see Table 4, d for scaling).

### 3.5. Sensitivity of the Topographic Analysis for DTM Uncertainties

The DTM differences at the pristine control sites where no man-made changes occurred represent the uncertainty connected with the proposed topographical method. Table 4a describes the SfM-derived elevation changes in Tammalampi and Kirkaslampi. The share of the significantly elevated and subsided areas inside the processing boundary was, on average, 77% smaller at the pristine sites. However, in those areas, the mean rise was only 20% smaller and the mean subsidence 40% smaller than at the restoration sites, on average. The pristine elevation changes were also more uniform, resulting in significantly lower STD values. Figure 6E reveals that most of the change inside the processing boundary of Tammalampi is related to the edges, where the accuracy is decreased towards the north and southwest due to poor SfM modelling. Despite the smaller changing area in the fully onboard-RTK-georeferenced dataset of Kirkaslampi (Figure 6F), the magnitude of the changes was similar. The differences north of the GCP OK3 are probably related to mapping errors caused by the tree cover. The changes east of OK9 are situated in treeless hollows containing slushy peat, which might swell significantly depending on the hydrological state.

Figure 8C,D shows how these minor elevation changes caused by uncertainties affected the flow accumulation and SWI values. The overall flow networks were similar

in both campaigns. At a smaller scale, detailed variation in the dispersion and location of the flow was recognized. However, the absolute changes in the main route lengths were significantly smaller (3.1–8.1%) than at the restoration sites (25–37%), but were still significant, particularly in the case of the flat terrain of Tammalampi. The changes in SWI were minor according to the histograms. The mean and STD absolute changes all remained smaller than at the restoration sites.

## 4. Discussion

### 4.1. UAS Mapping and SfM Processing Experiences

This paper introduced a study using UAS-based remote sensing supported by openly available LiDAR data to produce information to support peatland restoration monitoring. The mapping was found to be successful by providing new knowledge on peatland structural changes after restoration operations in ultra-high spatial resolution. In general, the mapping extents cannot typically cover whole restoration sites. Power consumption and battery capacity remain the primary drawbacks of the current technology, however, development of this technology is rapid. Mapping can also become more area-efficient when using fixed-wing unmanned aircraft instead of the multi-copters used in this study [24]. Detailed UAS monitoring should be focused on the areas of interest predicted to face significant changes in restoration or areas that can be used to represent the site as a whole.

Data intended for topographical analyses requires sufficient internal accuracy to create high-quality elevation models, as well as sufficient external accuracy when change detection is the focus of interest [71]. To achieve centimetre-level accuracy, the georeferencing must be based on precise surveying, such as RTK GNSS, of the aircraft location and/or GCPs [62]. Having enough tie-points between the neighboring images with sufficient overlap, is another key criterion [71]. On average, each of our tie-points had 5.7 projections. The modelled ground surface under dense tree canopy gave poor accuracy, and thus we cannot recommend SfM for mapping the grounds in such environments ([18] reported similar findings). The points in the open areas were produced from at least ten images, but the ground points under trees were typically based on only two. Acquiring data during the deciduous season has been suggested for increasing accuracy under tree cover [79], however, this was not applicable for our sites as they mostly consisted of coniferous trees. As well as overlap, UAS mapping accuracy has been shown to depend on altitude, number of GCPs, surface morphology, weather, platform, sensor and post-processing tools [23]. All these factors need to be controlled and the accuracy needs to be tested if the products are intended to be used for spatially advanced applications.

Permanent GCPs attached to the underlying mineral soil (Figure 9C1,2) were found to be a good option for intensive study sites under repetitive monitoring. The GCPs attached to the unstable organic soil were observed to move vertically during restoration. The roots of the utilized trees likely do not penetrate the prevailing water table level and thus, the stumps remain floating close to the surface [80]. Such a fine change might not be visible in the georeferencing residuals so cannot be traced internally. We strongly encourage UAS practitioners working in peatlands to always attach the GCPs to stable mineral soil or to survey their locations for each campaign. There were slight difficulties in pinning the precise marker centre in Metashape for part of the GCPs in the Mujejärvi IA/CA campaigns due to the 50 cm square aluminium plate on top of the wooden cross needed for simultaneous thermal mapping. Similar challenges were also found for OI6 (IB) and OK8 (CA) due to loosened screws causing rotation of the boards away from the perpendicular orientation. Furthermore, the 20 mm thickness of the GCP boards caused them not to align on the same elevation, giving an error on the visually determined cross centre. However, the errors had no effect when the GCPs were mapped from all directions, and we consider the errors not to be significant compared with the other uncertainties.

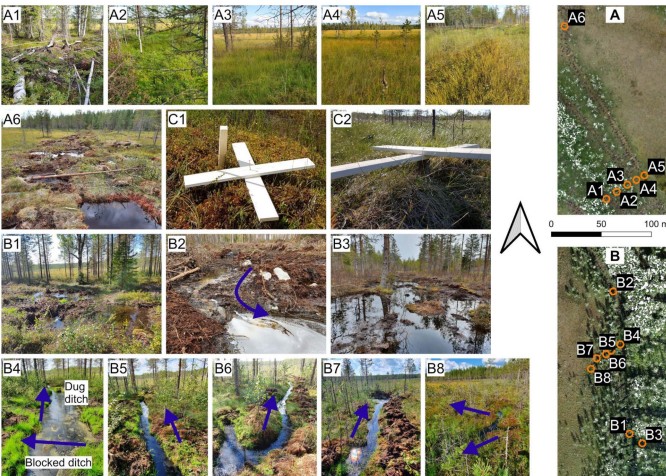

**Figure 9.** Field photographs and their shooting locations for the restoration sites of Loukkusuo (**A**) and Iso Leväniemi (**B**). The sequence (**A1**–**A5**) shows how the peatland surface develops when moving from the tree-covered area to the open peatland. (**A6**) and (**B1**) show the excavation footprint. (**B2**) shows flow-derived erosion on the ditch infilled with organic and mineral soil. (**B3**) shows the wide inundation during the high-water season after restoration. The sequence (**B4**–**B8**) shows how the water from the infilled ditch is directed to the open peatland through a manually-dug ditch. (**C1**) shows a GCP not yet attached to the pole hit through the peat and (**C2**) shows a GCP attached to a tree stump. The blue arrows represent the direction of water flow.

Some of the GCPs should always be reserved as checkpoints for testing the quality of the photogrammetric model. In Iso Leväniemi, the loose attachment forced us to choose the reference points with poor linear geometry. This seemed, however, not to affect the overall accuracy when the onboard-RTK co-ordinates were weighed as much as it did during the georeferencing. A linear GCP network would have been fatal for the non-RTK mapping, as it is known to be easily distorted (known as the doming effect [71]). For the non-RTK campaigns, we could only afford one checkpoint as there were nine GCPs in total per site. Still, we noticed some systematic errors that suggest insufficient GCP distribution [71]. In Tammalampi, the GCP network was rather narrow due to the dense tree cover surrounding the elongated open peatland. In Loukkusuo, it seemed that ML2 would have been needed as a reference point in the non-RTK campaign to split the great distance (191 m) between ML1 and ML3 in half. We strongly recommend a higher number of GCPs. Fifteen usually suffices (see [23]) to ensure enough GCPs for georeferencing, as this number allows several to be used as checkpoints and leaves a safety margin for production failures. Some studies have shown eight reference points suffice [71], but a larger number enhances the optimization of the camera parameters [62]. For high-quality elevation data needs, we recommend using onboard RTK together with five GCPs distributed as a quincunx. Even with fewer GCPs, a precise GNSS unit and a walk around the site are still required, so surveying some extra checkpoints is an easy way to secure product quality. If the ground control is not an option, the error of the onboard-RTK can also be decreased by adding oblique imagery to the dataset [81,82]. An external control dataset is then recommended. Including imagery from several altitudes permits the precise calibration of the camera parameters, especially for flat terrains [83]. SfM-based point clouds typically have a high noise level and outliers exist, e.g., due to complex geometric structures and reflections [68]; this is especially true for vegetated sites due to vegetation being moved by wind [62]. A discussion of the used filters can be found in Section S4 of the Supplementary material.

The SfM processing procedure included many steps and plenty of parametrization (see e.g., [84,85]). While many parameters were able to be justified based on the literature and experimenting, several remained arbitrary. Metashape has its own user community sharing experiences on an online forum [86], but straightforward guidelines would be needed for restoration practitioners who do not necessarily have time to specialize in all steps, such

as those in the optimization and gradual selection tools. Furthermore, sometimes even researchers leave relevant parameters as defaults due to the absence of better knowledge. For instance, we noticed that setting the assumed accuracy of the GCP co-ordinates (often left as the software default of 5 mm) to a more realistic 20 mm significantly increased the produced RMSEs.

### 4.2. Topographical Aspects in Peatland Restoration

According to the results, the primary changes with dimensions up to 1.0 m, such as ditch infilling, dam and embankment construction and the related excavation pits is shown within the data accuracy level. Furthermore, we showed minor secondary changes which needed expertise in interpretation to be elaborated from the mapping errors and natural surface variation. The presented flow accumulation and wetness maps showed that the flow path development was mostly related to the major primary changes, as the secondary and the error-related changes in the study area were minor. In the degraded state, the ditches guided the flow to straight routes following the ditch network, whereas the routes got convoluted and dispersed after restoration. The redistribution of flow and wetness seemed to depend on the relative orientation of the drainage network. In Loukkusuo, where the ditches were perpendicular to the general hydraulic gradient, the flows seemed to pass the infilled ditch lines without interruptions, achieving a pristine-like dispersion. In Iso Leväniemi, most ditches were parallel with the general hydraulic gradient and the change in the routes was milder, but the flow still got convoluted, promoting wetness, likely decreasing flow rates and preventing erosion. Logically, the wetness was decreased in the earlier ditch bottoms and increased in the areas that had previously been effectively drained.

Restoration was assessed as successful according to the topographical analysis, since the flows dispersed, the overall wetness increased and the sites became more evenly wet. Furthermore, the total length of the flow paths increased, which is the opposite of what is known to happen after drainage [53]. However, the excavation pits in Iso Leväniemi might have preserved the flows close to the infilled ditches. UAS mapping showed the potential for documenting the pit locations (the personal excavation footprint of the driver, Figure 9A6,B1) and the possible undesired interconnectivity. The high slope was a challenge for dam construction in Iso Leväniemi. Wetness could have been promoted more by building higher and longer embankments at a lower interval to achieve a sufficient flow dispersion in the strips between the ditches. It is known that the infilled ditches are sensitive to gather flow even after the restoration since the drained organic soil encounters subsidence, lowering the land surface close to the ditch [12,17]; this was observed in Iso Leväniemi. Furthermore, the ditches were deep, making it difficult to gather enough infilling material without lowering the ditch surroundings. The parallel ditch orientation seemed to emphasize the flow route permanency.

The implementation of the Loukkusuo site included the construction of dams. Our results question their viability as, according to the topographic information, the perpendicular ditch orientation (i.e., a parallel dam orientation) barely impacted the flow. However, the ditches might still gather the flow if the hydraulic conductivity is higher for the filling material than for the surrounding peatland. Furthermore, the degradation rate of the peat increases when lifted to drier, aerated conditions. The dams being higher than their surroundings allow for future subsidence of the restoration constructions and might also be useful during the high-water season. The parallel and the perpendicular examples are the extreme cases. Usually, ditch orientation is between these two extremes. It is also important to document whether excavation was able to be completely conducted. In Loukkusuo, the wet slushy soil could not hold the excavator during the summer implementation (Figure 7A2). However, the related ditch location was not visibly operational before the restoration due to the overgrown vegetation and the ditch leaked to the non-ditched area according to the flow accumulation and orthomosaic (Figure 8A).

Erosion caused by rapid surface flows can also be a significant reason for technical failures in peatland restoration. No impacts of erosion were spotted with the Iso Lev-

äniemi UAS data despite the high-water season also including the highest flow rates in the monitoring period. Some channelized flow in the infilled ditch at the north-eastern edge of the undrained part was seen in field monitoring in May (Figure 9B2), but the issue was repaired through manually dug ditches and embankments which directed the flow to the open peatland before the IA UAS campaign (Figure 9B4–B8). However, the ditches were too narrow to be considered in the topographical analysis. Dale et al. (2020) performed a multitemporal UAS SfM study at an intertidal saltmarsh under managed realignment [19]. Their morphological change included erosion and accretion of sediment that caused significant impacts on the drainage network.

Minor secondary elevation changes were considered to be caused by changes in the peat wetness, but the comparisons were disturbed by natural surface variation and mapping errors. Peat volume is known to vary according to the water content [87]. Rapid increases in water table levels have been reported after peatland restoration [10,11]. However, the time between the restoration and the mapping campaign was only a few weeks in Loukkusuo, which seems too short for enough water to be gathered for swelling to take place, particularly in the middle of the driest season. Instead, the ditch infilling seemed to immediately restrict the discharge through an earlier main route (Figure 8A), since the decrease in the wetness would explain the shown subsidence in the area (despite the varying accuracy). The twelve-month monitoring period in Iso Leväniemi was enough to produce swelling. Interestingly, the swelling was strongest at the lowest surfaces, i.e., the flarks (Figure 7B3). On boreal fens, the flarks are typically covered with the wettest plant communities shown to be prone to elevation changes [87]. They form a loose soil structure and might, thus, be more sensitive to swelling than the higher strings strengthened with vascular roots. However, swelling of flarks also existed in pristine Kirkaslampi. A longer time series would help verify whether the swelling was due to the restoration or natural variation in wetness.

### 4.3. Observations from the Control Data

The quality inspections revealed the UAS SfM mapping to achieve a <100 mm spatial accuracy for the data used in the topographical analysis. The lowest mean $RMSE_Z$ of 16 mm was determined for the checkpoint GCPs. The GCP crosses are straightforward for the SfM algorithm as they are stable, clear-shaped objects, in contrast with the ambiguous peatland surface. In addition, GCPs only represent discrete locations, while the quality of the data between remains unknown [71]. Thus, reference data from the peatland surface was also used for model evaluation. Field-measured elevations from the instant of the UAS mapping campaign are recommended as reference data, but those were not available for this study. A cell-wise comparison of two pristine SfM DTMs better represents the accuracy of the analysis data. The cell-wise mean $RMSE_Z$ of 48 mm fits well into the typical range of 40–60 mm (review of fifty SfM studies by [23]). For dense tree cover and the non-RTK campaigns, distance to the GCP network and poor image overlap lowered the accuracy. Thus, it is recommended to always map a safety margin and crop the edges afterwards. We chose the location of the processing boundaries rather arbitrarily. Some poorer accuracy zones also needed to be included for data integrity and continuity. The Level of Detection (LoD) approach was used to consider the accumulating DTM errors when drawing elevation comparisons. The DTM was able to describe sub-decimetre elevation differences when LoD was determined using the checkpoint errors (similar to [19,71]). Considering the cell-wise errors inside the processing boundaries doubled the mean LoD, the LiDAR LoD was not considered representative as the sensor-related bias is pronounced. Furthermore, LoD was found to be only half-applicable for approaches where a single error is produced, e.g., the error is determined as the elevation difference between two campaigns or there is no control data for the other campaign, such as in this study.

A major issue with the SfM in vegetated environments is its tendency to model the ground surface into the canopy of any prevailing dense understory vegetation, such as sedges or dwarf shrubs (Figure 9A1–A5), whereas LiDAR technology typically measures

points through the vegetation, coming closer to the so-called surface of the vegetated peatland (*Sphagnum* canopy in many cases) [22]. However, LiDAR accuracies also decrease under dense vegetation [36]. We noticed the LiDAR systematically producing several centimetre lower elevations than the SfM, with this being even lower at the tree-covered parts. Reference [26] compared an SfM-derived DSM and a terrestrial laser scan, concluding that vertical planes were especially challenging for the SfM. Thus, any merging of data from different mapping sensors should be interpreted with caution as they describe the terrain differently. Preferably, the mapping campaigns should be implemented using consistent mapping technology. Despite the discrepancy, the LiDAR data was, due to its ubiquity and external production, overpowering in the comprehensive accuracy assessment of the SfM data. The mean $RMSE_Z$ of 98 mm determined using LiDAR data indicates the dataset accuracy to suffice for revealing the primary topographical changes of 0.6–1 m.

The impacts of natural variation and errors due to mapping and processing were shown with multi-temporal datasets from the pristine control sites, where no excavator operation occurred. The open parts of the pristine sites included very little elevation development (Figure 6E,F). The season strongly affects the hydrological conditions in the boreal zone. At our sites, the water tables are typically highest at the end of the thawing season and lowest in the late summer. Correspondingly, the volume of the peat varies according to its water content [87]. Simultaneously, the 3D structure of the interpreted peatland surface altered according to the developing phenology of the annual understory vegetation (Figure 9A1–A5, [22]). For instance, the middle part of Loukkusuo is dominated by grasses that might have increased the surface elevation during the observation period from June to August. However, the area showed subsidence that would have been logically consistent with the decreasing wetness, but still might be related to the poor non-RTK georeferencing of the IB UAS dataset. The best solution for excluding the impact of natural variation is to implement the UAS campaigns in hydrologically and phenologically identical seasons. In general, the flow accumulation at the pristine sites seemed rather stable against the errors in the DTM (e.g., [21] reported similar), except for the most even surfaces.

Even with the mentioned uncertainties, we showed external evidence supporting the results of the topographical analysis. Signs of increased wetness were recognized in orthomosaics. In Iso Leväniemi where the hill promotes groundwater discharge, a fresh seepage surface was noticed in the corresponding location, where the flow accumulation predicted a significant new flow route after the restoration (Figure 8B). Furthermore, we tested the predicted wetness with the sampled soil water content (SWC) that showed a statistically significant correlation with the restoration sites, arguing for realistic predictions. However, the modest coefficients of determination achieved ($R^2$ = 0.26–0.42) suggest that field wetness also depends on other factors besides topography. Similar findings have been reported in the literature. Riihimäki et al. (2021) showed correlations of up to $R^2$ = 0.23 between SWI and soil moisture measured with a reflectometry sensor in the arctic tundra [37]. At the same site, Kemppinen et al. (2018) achieved a predictive performance of $R^2$ = 0.47 with statistical models while including other factors, but concluded that SWI was the strongest predictor [46]. Controversially, Kopecky et al. (2021) measured soil moisture using microclimate loggers in a temperate forest and SWI was explained rather poorly with an $R^2$ = 0.13 [88].

Comparing the differences between our site types, we suspect the failure in fitting the pristine regression models to be due to the small catchments, i.e., a certain upslope area threshold is required for the topographical model to function properly. On the other hand, the correlations might be influenced by site specific subsurface processes [88]. In pristine Kirkaslampi, the determined upslope catchment was almost as small as the processing boundary due to it being confined by an upslope land ridge. However, groundwater inputs from the hill are known to discharge at the site as seepage surfaces. The depression behind the ridge was not determined as a topographical sink, however, filling it would have permitted a more realistic catchment for the site. Similar groundwater inputs were present in well-correlated Iso Leväniemi, but for that site, the modelled flow origin reached

the hilltop (Figure 1). Even so, the small catchment modelled for pristine Tammalampi represented the true water input to the site well, i.e., the poor correlation argues for a minimum upslope area. We consider our SWC datasets which were acquired after the restoration to be rather limited and recommend further ground validation studies, including larger datasets for both states.

### 4.4. Limitations of Topographical Analysis

Topographic wetness indices (TWIs) assume that the local slope acts as a proxy for the hydraulic gradient [88]. Single-flow algorithms direct complete flow packages from one cell to another. They typically produce sharp wetness distributions, resulting in neat flow accumulation maps which are not ecologically plausible [89,90]. They also tend to produce artificial patterns due to the flow constrained to a regular sampling grid [45,91]. The use of multiple-flow algorithms can divide a flow package into several downslope cells, which has been shown to produce more realistic, dispersed flow and wetness predictions [37,44,45,89]. They reduce the bias typical for flat sites, i.e., for flow routes that are more sensitive to variation 88,41,72,74,90]. We used an adapted TWI called SAGA Wetness Index (SWI) which better represents the high water tables typical of flat areas [75]. SWI uses the multiple-flow algorithm developed by [76] and iterates the SCA among the neighboring cells. If the flow is focused on a specific route on the flat surface (multiple-flow algorithm should already have prevented this), all cells with a similar slope nearby are considered to be wet, despite not being situated directly along the route [88].

The impacts of resolution on topographical analysis have been widely studied (e.g., [36,37,39,40,44,47]). Coarser resolutions describe the microtopography inaccurately and typically result in smaller slopes, larger upslope areas, less flow divergence and shorter flow paths, and thus, larger values of the wetness indices [36,37,39,44]. The desired resolution also depends on site specific terrain features [40,46,72] and the hydrologic response willing to be observed. Fine-scale microtopography correlates with soil moisture content but a coarser topography drives groundwater flows [36,41,47]. We chose to resample to 1 m before the topographical analysis as this study aimed to describe the flow and the general wetness patterns adequately but not to reveal the wetness differences, e.g., between the strings and the flarks. Coarser resolutions would have considered the sites as flatter, likely causing flow overdispersion and crucial errors in the routing [37,41]. Riihimäki et al. (2021) showed the SAGA algorithm performing best at a resolution of $\leq 2$ m, while coarser data was optimal for most of the other algorithms [37]. However, resolutions of <1 m have been found to be unsuitable for topographic analysis despite thoroughly describing the microtopography. They produce smaller micro-basins and more varying slopes, resulting in irregular pathways without proper connectivity [39,41,44]. However, with the used resolution, details relevant to peatland restoration, such as small-scale erosion and narrow ditches, remain hidden.

Despite TWIs representing both surface and shallow sub-surface flows [36,38,39], proper consideration of the groundwater interactions remains one of the main challenges. In Olvassuo in particular, the groundwater routes are complex, and the study sites receive local groundwater at the upper stages and regional groundwater at the lower stages [92]. The upslope area's contribution to flow depends on how much water is infiltrated and if the infiltrated share continues following the slope or moves down to the deep routes and later back toward the surface. Furthermore, the infiltration and the sub-surface flow paths and rates strongly depend on peat depth [46] and decay grade [35], as well as on the permeability of the surrounding and underlying mineral soil [88]. To describe such diverse flows and consider their impacts on the flow path analysis, extensive 3D data of the soil hydraulic properties would be required. Another feature of organic soils is their strong positive feedback on soil moisture. Peatlands originally appeared on lowlands disposed to wetness and, in a pristine state, they hold high water contents, typically having saturated conditions that are determinative for the flow production [46]. It is also known that the hydraulic gradients might be smaller than the corresponding slopes at the edges

of wetlands due to the high water tables downslope [41]. This might produce bias in the topographical analysis. Furthermore, drainage dramatically changes the hydrological properties of the peatland [87], which might be one reason among other uncertainties why we encountered different correlations for the degraded and pristine sites. Our sites were minerotrophic fens that typically receive their water inputs based on the topography. Controversially, a typical topography of ombrotrophic bogs isolates the peatland from any input flows due to the surface being higher than the surroundings. In this case, the peat's ability to hold water might be more important than the topography and thus, TWI assumptions are not necessarily met. However, Lendzioch et al. (2021) showed SWI to be an important predictor of groundwater level on a temperate mountain bog [93].

The sites were affected, not only by the precipitation inside the UAS-mapped area, but also by the water inputs from the catchment upslope that needed to be considered for comprehensive results [74,88]). Acquiring UAS DTMs for the upslope catchments was impossible due to their large area and dense tree cover. A ubiquitous elevation model was thus required to be fused with the SfM-based DTM. However, high-resolution ubiquitous data is not globally available. How low the resolution can be while still sufficiently describing the water input is something to be tested. UAS LiDAR (e.g., [22,91,94]) could be another option to map the tree-covered upslope catchments, at least in the size class shown in Tammalampi, as it would also solve the challenges with data merging. The largest catchments of >1 km$^2$ are not an option for unmanned solutions. Furthermore, LiDAR technology is still expensive compared with affordable photogrammetric mapping.

On the other hand, a high resolution might include details which disturb the results. Since our ubiquitous data were acquired in the IB state, bias remained in the upslope flow routes. Dissolving the upslope ditches computationally would cause elevation conflicts at the data boundaries [95]. Some conflict already existed, while the ditches suddenly became infilled at the boundary, but the sink filling seemed to adequately handle such transformations. We recommend further studies on sink processing in peatland restoration by comparing the filling and breaching methods. Breaching has been shown to be efficient, particularly when man-made constructions such as culverts underneath roads exist, and particularly in flat environments [77]. It would be essential to understand how sink filling and breaching affects studying the interconnectivity of the excavation pits. Breaching might also create a potential for modelling the groundwater passage through the perpendicular mineral ridges.

Flow accumulation and TWIs should be considered as predictors only and they should be co-interpreted with the topographical changes [19]. The mass balance assumption in the TWI approach requires constant hydrological conditions during the observation period, i.e., uniform precipitation producing constant infiltration and flow through a uniform transmissivity for the whole area analyzed [74,90]. Furthermore, evaporation depending on the locally varying exposure greatly affects the true SWC [89]. Thus, the predicted wetness can be considered as a long-term average rather than conditions at an actual moment [41]. Particularly mineral soils and shallow peat soils encounter large seasonal variation due to their low water-holding capacity, while deep peat soils have more stable conditions and better resistance to evaporation [46,93]. Our field sampling timing simultaneously with the UAS campaigns might not be optimal. Riihimäki et al. (2011) used TWIs to explain soil moisture as being poorest in August and best in June, i.e., the optimal timing seems to be at the beginning of the growing season when the water tables are still high [37]. If there are high-resolution meteorological data available, advanced model-based wetness indices might describe the hydrological dynamics in greater detail than the static TWIs [41]. Other approaches for flow accumulation and topographic wetness are recommended for further research, e.g., the depth-to-water index [40], topographic openness [96], Facet-Flow Network [97], and triangular facet network [98]. However, we argue that the presented approach, as simplified as it is, is sufficient for revealing the main hydrological impacts of primary elevation changes in peatland restoration. As the topographical analysis in ultra-high resolutions is a rather fresh research topic, many details still need attention.

*4.5. Management Implications and Wider Applicability*

The primary aim of peatland restoration is a spatially comprehensive rewetting of the site for the recovery of pristine-like processes such as carbon sequestration, succession toward biodiversity and retaining the natural function in the catchment hydrology. As peatland degradation has been more and more understood as a global threat, larger areas become subject to restoration each year, and novel, area efficient and harmonious monitoring methods are needed [13]. Monitoring aims at evaluating the success of the technical implementation to anticipate corrective actions and develop restoration methodology [9]. Conventional peatland monitoring is based on laborious ground observations. Elevation surveys have been conducted with level, total station and RTK GNSS (e.g., [16]). The first field monitoring visits after restoration typically include subjective visual assessment of factors such as amounts of incoming water, nature-like wetness distribution and effectiveness of ditch infilling and dams in terms of water retention [9]. Furthermore, hydrological changes have been monitored from standpipe wells, either manually or using devices automatically logging the water levels or soil moisture in high temporal resolution [10–12]. However, these sensors are restricted to discrete locations. To support and supplement the conventional approaches, we showed a methodology that can help to spatially estimate the hydrological impacts of restoration. Remote sensing provides opportunities for representative unbiased inspections by gathering data efficiently from large areas. UAS SfM is a cost-efficient method for producing 3D data at ultra-high resolution. However, it is not applicable for areas covered by dense vegetation or which extent over more than a few dozen hectares [23].

It is not clear whether mean wetness would increase due to restoration. The same water input could be redistributed to a wider extent, including more routes with less water on each. We have provided evidence of the bidirectional changes in Loukkusuo, where the surroundings of the earlier main flow routes became drier while new routes increased the surrounding wetness. Despite the upslope catchment area remaining constant, the slopes became milder due to the ditch infillings and thus, the tendency of the water to be discharged decreased. Even if the TWIs are not directly applicable to larger extents or comparable between sites nationally [90], they provide great potential for deriving site-specific before and after comparisons when temporal elevation data is available. It also seems that the development of TWIs and flow accumulation cannot be directly compared with pristine sites such as water tables [11] even if the restorations caused some of the values to approach pristine ones. The total length of the main flow routes, in particular, showed no clear trends, despite the varying site areas being considered. However, the pristine sites showed their applicability in evaluating the sensitivity of the method for external factors, such as mapping errors and natural variations in the surface. We encourage researchers to map pristine control sites simultaneously with the experiment sites for a full understanding of the experimented phenomena. For topographical analysis, the control site selection should not be based on vegetation type only, but also on the similar slope characteristics and upslope catchment area.

Our method could also be used to assess the hydrological impacts of restoration projects where the peatland site has not been drained directly, but instead the ditches in the upslope catchment have directed the flows past the site, causing secondary drainage. In these cases, a few directing ditches (such as in Figure 9B4–B8) from the collecting ditch above might be sufficient for rewetting the peatland, but determining the impacted area while meeting the interests of funders and policymakers is challenging as the implementation area remains small. Flow accumulation has the potential to quantify the major downslope impacts of these minor operations. When a ubiquitous DTM is available, it could be used as an elementary model that can be replaced by the detailed UAS data for the change areas only, e.g., the surroundings of the directing ditches. Similarly, for restorations including ditch infilling and damming, UAS mapping could be focused on the ditch lines if the area does not support a grid-type flight plan. Excavation is typically anticipated by a clear-cut along the planned paths for excavators [9], and these linear harvest openings

seem to be wide enough for UAS mapping. Furthermore, topographical analysis has great potential for restoration planning to predict hydrological effects and the planning of the position and size of dams. We encourage the scientific community to further continue developing this methodology.

## 5. Conclusions

This paper tested and demonstrated UAS-SfM-derived DTMs supported by ubiquitous LiDAR data as inputs to a topographical analysis producing maps of flow accumulation and wetness to assess the restoration of two boreal minerotrophic peatlands. UAS mapping showed the potential in mapping the ground elevation changes in ultra-high spatial resolution. While the mapping process is effortless and highly automated, the data quality needs to be ensured with precise georeferencing, sufficient overlap between the neighboring images and the use of external control data. The accuracy was shown to be sufficient for evaluating the topographical impacts of major primary changes such as ditch infilling and dam construction. The restorations were considered successful due to the increased mean wetness and the narrowed wetness scale. In addition, the flow was dispersed particularly when the ditches were perpendicular to the general hydraulic gradient. We also showed weaknesses of the used restoration measures such as the absence of dams in Iso Leväniemi and the futility of dams parallel to the general hydraulic gradient in Loukkusuo. The presented topographical method provides a novel type of spatial understanding of the peatland restoration impacts and offers great potential for the increasing monitoring needs.

**Supplementary Materials:** The following supporting information can be downloaded at: https://www.mdpi.com/article/10.3390/rs14133169/s1, Table S1: Development of checkpoint RMSE, reprojection error and number of removed points after filtering the sparse cloud and the sub-sequential optimization of the cameras; Section S1: Metashape Filter Parametrization; Section S2: Statistical Outlier Filter Parametrization; S3. Cloth Simulation Filter Parametrization; Section S4. Filter discussion; Section S5: Correlation between Topographic and Field Wetness; Figure S1: Gravimetric Soil Water Content (SWC) as a function of topography-based Saga Wetness Index (SWI) and the fitted regression lines. References [99–111] are cited in the Supplementary Materials.

**Author Contributions:** Conceptualization, L.I., A.-K.R., H.M. and J.I.; Methodology, L.I.; Software, L.I.; Validation, L.I.; Formal Analysis, L.I.; Investigation, L.I., M.S., S.R. and L.P.; Resources, L.I., H.M. and T.K., Data Curation, L.I., T.K., M.S. and L.P., Writing—Original Draft Preparation, L.I., Writing—Review & Editing, L.I., H.M., A.-K.R., M.S., S.R. and L.P.; Visualization, L.I.; Supervision, H.M., A.-K.R., J.I. and B.K.; Project Administration, A.-K.R., H.M., J.I., B.K., L.I. and T.K.; Funding Acquisition, A.-K.R., H.M., B.K. and T.K. All authors have read and agreed to the published version of the manuscript.

**Funding:** This work was supported by the Hydrology LIFE project by European Union LIFE Programme: LIFE16 NAT/FI/000 583; WaterJPI WaterWorks2017 ERA-NET Cofund project WaterPeat (project number 326848). The writing was supported by Hydro-RDI-Network (grant numbers 337280 and 337523) by the Academy of Finland no. 337280 and Maa- ja Vesitekniikan tuki ry no. 14-8844-22. T.K. was funded by the Strategic Research Council (SRC) decision no. 312636 (IBC-Carbon) and by Academy of Finland decision no. 347862 (C-NEUT).

**Institutional Review Board Statement:** Not applicable.

**Informed Consent Statement:** Not applicable.

**Data Availability Statement:** Publicly available LiDAR datasets were analyzed in this study. These data can be found here: https://tiedostopalvelu.maanmittauslaitos.fi/tp/kartta?lang=en (accessed on 8 November 2021). The UAS data presented in this study are available on request from the corresponding author. The data are not publicly available due to being included in other publications in preparation.

**Acknowledgments:** Pasi Korpelainen (Dronelab, University of Eastern Finland) collected the drone data used in this article. Sirkku Ahonen (Lapland University of Applied Sciences) performed pre-analyses for the SAGA GIS.

**Conflicts of Interest:** The authors declare no conflict of interest. The sponsors had no role in the design, execution, interpretation or writing of the study.

## Abbreviations

| | |
|---|---|
| 3D | Three-dimensional |
| CA | Control After (state at the control site after the restoration at the intervention site) |
| CB | Control Before (state at the control site before the restoration at the intervention site) |
| CSF | Cloth Simulation Filter |
| DEM | Digital Elevation Model |
| DSM | Digital Surface Model |
| DTM | Digital Terrain Model |
| GCP | Ground Control Point |
| GNSS | Global Navigation Satellite System |
| IA | Intervention After (state at the intervention site after the restoration) |
| IB | Intervention Before (state at the intervention site before the restoration) |
| LiDAR | Light Detection and Ranging |
| NLS | National Land Survey of Finland |
| RMSE | Root Mean Square Error |
| RTK | Real-Time Kinematic |
| SfM | Structure-from-Motion |
| SWC | Soil Water Content |
| SWI | Saga Wetness Index |
| STD | Standard Deviation |
| SOR | Statistical Outlier Removal |
| TWI | Topographic Wetness Index |
| UAS | Unmanned Aircraft System |

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
