# Peer review of "Unmanned Aircraft System (UAS) Structure-From-Motion (SfM) for Monitoring the Changed Flow Paths and Wetness in Minerotrophic Peatland Restoration"

_remotesensing, doi:10.3390/rs14133169_

Round 1

Reviewer 1 Report

This manuscript was very well written, thorough, detailed, and clear. It presents a UAS Structure-from-Motion method for measuring changes in flow paths and wetness from Peatland restoration. I do have a few points below with suggestions for clarifications or added discussion about the method:

- As a drone pilot myself I am curious what autopilot software was used for the mapping, and some deails on the mapping (mapping of an area made during one day or spread out over several days with many flights?).

-  Using a Phantom 4 Pro/RTK drone for cm-level mapping is a potentially fast and cost-effective way of making a surface model and calculating the wetness. The proposed method needs support from ubiquitious LiDAR data (e.g. L455-456: "The ubiquitous 2 m LiDAR DTM from the Deg/CDeg state was used for estimating the water input from the catchment upslope") which may have be done seldom or lacking in a database. As a reader I would like to see a paragraph discussing future potential improvements or variations of the method, e.g. would a drone equipped with a LiDAR+RGB camera (such as the DJI Matrice 300 RTK equipped with a DJI Zenmuse L1 LiDAR/RGB) be a more efficient method to use? more accurate for the wetness calculations as LiDAR data is produced by the drone at custom resolution over any area and time period? Could it be a more automatic/easy to process method for novice users? Also not requiring merging of datasets from different sensors (SfM and LiDAR)?

- There is no mention of a RTK base station for the on-board RTK flights (e.g. D-RTK2 Base Station ). From my understanding this is needed for the RTK system to give cm to dm level accuracy (instead of the meter-level accuracy of non-RTK flights at high altitude which relies solely on GPS).

- Figure 3, difficult for the reader to get a grip on what areas these orthomosaic pictures correspond to in Figure 2. Could e.g. draw a red square in Fig.2. marking the boundaries of the Fig.3. orthomosaic pictures. When reading the manuscript I found myself going back in forth between Fig's 2 and 3 trying to find the scales and boundaries involved in Fig.3.

Minor things:
+ "Figure 1." in the caption for Fig.1. is not bold and in another font?
+ L703. "this seemed" -> "This seemed"
+ Figure 5. No verical axis is shown. Would be helpful for the reader to know the vertical scale (to approximate the deviations in the y-axis). There scale is shown at the bottom of the Fig. but is this the same for both the x and y-axis? (in that case this should be stated in the Fig. caption).

+ Table 2. "Number of aligned cameras", what is mean by this? Is this the number of images = camera positions?

Reviewer 2 Report

The article titled UAS structure from motion for monitoring the changed flow paths and wetness in minerotrophic peatland restoration aims to assess the success of peatland restoration in two sites in Finland using DTMs derived from UAS footage of the sites before and after restoration. The experimental design included a pristine control site for each of the restoration sites. I was unsure whether the main aim of the paper was to assess the methodology of using the UAS data to determine changes in elevation and flowpaths before and after restoration - and to compare these with other monitoring methods; or whether the methodology is considered established and it is the changes brought about by restoration that are the main focus of the work. The aim should be more clearly highlighted for readers.

Overall the paper makes a novel contribution to the research area to provide a method to assess the spatial scale of the impacts of peatland restoration on site hydrology and hence give an idea of the effectiveness of restoration. It would be interesting to know if the size of the elevation differences continues as the absolute changes seem to be high in some cases compared to elevation changes we have measured on UK raised bogs (have seen changes of 2-3 cm post restoration, as measured by in-situ monitoring, which is a factor of 10 lower than seen here. A weakness of the methodology is that there don't appear to be any in-situ measurements of peat surface elevation change.

I feel that the manuscript is generally good but could benefit from some rethinking in set-up to aid the reader. The first is to clarify the main aim of the work (as mentioned above), the second would be to have a clearer way of labelling the sites in text, tables and figures. My understanding of the method is that it is a before/after, control/intervention design but I struggled to follow this at first because the control sites were labelled "control degraded" and "control restored" and I was trying work out what restoration had occurred in the control sites. Maybe label them as "control before", "control after", "intervention before", "intervention after". Also, probably due to not being a Finnish speaker, I struggled to remember the names of the sites so had to keep cross-referencing to work out which was which. A minor correction that would help that is if the sites mentioned first in the text were on the left hand side of the associated figures and tables.

Specific comments include:

* Title - spell out UAS

* line 106: detailed not detail

* Table 2 (and others): make sure dates aren't squashed over two lines

* Table 2 (and others): clarify journal formatting for large numbers, I was unsure whether for Loukkusuo the number of tie points was 269 and 289 or 269289 (I would write as 269,289)

* Figure 2: can the processing boundaries be added to the images?

* Figure 7: check use of commas vs decimal points in figure legend (in tables and text decimal points are used).

* Fig 7: I would use false, not fallacious (and subsidence not subsidense)

* Fig 7: is this based on UAV data or the Lidar data? I got a bit confused. If different data to Fig 6 changing the colour in the figure may help the reader.

* Fig 8: what do you mean by benches?

* Line 1026: visual would be better than ocular in this context.
